# Number and dissimilarity of global change factors influences soil properties and functions

Mohan Bi[1,2,5], Huiying Li[1,2,5], Peter Meidl[1,2], Yanjie Zhu[1,2], Masahiro Ryo [3,4] & Matthias C. Rillig [1,2]✉

Soil biota and functions are impacted by various anthropogenic stressors, including climate change, chemical pollution or microplastics. These stressors do not occur in isolation, and soil properties and functions appear to be directionally driven by the number of global change factors acting simultaneously. Building on this insight, we here hypothesize that co-acting factors with more diverse effect mechanisms, or higher dissimilarity, have greater impacts on soil properties and functions. We created a factor pool of 12 factors and calculated dissimilarity indices of randomly-chosen co-acting factors based on the measured responses of soil properties and functions to the single factors. Results show that not only was the number of factors important, but factor dissimilarity was also key for predicting factor joint effects. By analyzing deviations of soil properties and functions from three null model predictions, we demonstrate that higher factor dissimilarity and a larger number of factors could drive larger deviations from null models and trigger more frequent occurrence of synergistic factor net interactions on soil functions (decomposition rate, cellulase, and β-glucosidase activity), which provides mechanistic insights for understanding high-dimensional effects of factors. Our work highlights the importance of considering factor similarity in future research on interacting factors.

Global change factors (GCFs) induced by human activities have a significant impact on soil physicochemical properties, process rates, and microbial communities in diverse terrestrial ecosystems[1]. The effect of individual GCFs on soil properties and functions have been the focus of many prior studies. For example, salinity reduces the availability of soil nutrients and has negative impacts on soil microbial activities[2], pesticides can pose adverse effects on non-target soil organisms[3], and drought affects soil processes by directly stressing soil organisms and indirectly by hindering substrate transport[4]. The multitude of GCFs collectively gives rise to concerns about soil ecosystem health.

Only a few studies have addressed the effects at a high-dimensional factor level with concurrent effects of a larger number of factors. A systematic mapping showed that fewer than 2% of experimental studies have explored the combined effects of three or more factors in the context of soils[1]. One of the main obstacles for studying joint effects of multiple factors at a time is the combinatorial explosion problem[5]. For traditional factorial designs, when the number of factors increases, the number of possible factor combinations will increase rapidly, meaning that such designs including a large number of factors are not feasible in ecology. To overcome this experimental challenge, recent studies investigating the effects of multiple factors

[1]Freie Universität Berlin, Institute of Biology, Berlin, Germany. [2]Berlin-Brandenburg Institute of Advanced Biodiversity Research (BBIB), Berlin, Germany. [3]Leibniz Centre for Agricultural Landscape Research (ZALF), Müncheberg, Germany. [4]Brandenburg University of Technology Cottbus–Senftenberg, Platz der Deutschen Einheit 1, Cottbus, Germany. [5]These authors contributed equally: Mohan Bi, Huiying Li. ✉e-mail: rillig@zedat.fu-berlin.de

followed an approach involving randomly selecting factors from a predefined factor pool; such a design avoids factor combination problems without losing generalizability[1,6,7]. Using this experimental approach addressed a general feature of multiple GCFs—the number of co-acting factors—which has been shown to directionally drive the effects of co-acting multiple factors on plants[8], soil ecosystems[1] and the plant community[6,9]. Another study also indicated that the increasing number of factors diminished the functions of soil microbial diversity[7]. However, in general, our knowledge about mechanisms underpinning the effects of multiple co-acting GCFs is still limited.

To gain more insights into this high-dimensionality problem, studies have highlighted the importance of ordering and classifying GCFs from trait-based perspectives[10–12]. In previous studies, factors are usually grouped by their sources instead of considering their effect mechanisms and ecological-scale dependency[12]. Recently, an a priori factor classification system has been introduced, using inherent traits (physical, chemical, and biological agents) and theoretical effects (effect mechanism, targets, and key properties) of 30 different anthropogenic factors[10]. Building more comprehensive factor classification systems may enable extracting features from factor traits to predict general patterns of multiple GCF effects. In this context, factor dissimilarity is a plausible feature that can be generated from multiple available factor traits and may capture patterns of high-dimensional effects[11]. However, the role of factor dissimilarity in driving the effects of multiple GCFs has never been investigated experimentally.

Another important feature of the multiple co-acting GCF effects is the nature of factor interactions. Across marine and terrestrial ecosystems, many studies found that when two or more factors are present, the combined effects often differ from what is expected based on the single factor effects[13–15]. The interactions between factors are defined as antagonistic when combined effects are less than expected, while synergistic interactions cause combined effects larger than expected effects. Although understanding interactions among factors is crucial for prioritizing ecosystem stressor management, when more than three factors are acting simultaneously, testing every pairwise factor interaction or high-order interaction is extremely challenging unless every factor combination has been separately replicated[16]. In this case, revealing the overall net interactive effects of multiple GCFs is a more practical solution and can also indicate potential interactions among multiple GCFs. Nevertheless, there is still a lack of established methods for identifying the net interactions for multiple co-acting GCFs and insufficient knowledge about the potential mechanisms underpinning such effects.

In this work, we aim to investigate the joint effects of multiple GCFs on soil properties and functions, examining the effects of number of factors and dissimilarity of GCF combinations. We present a microcosm experiment (Fig. 1) with 2, 5, and 8 factor levels to test the following hypotheses: (i) factor dissimilarity can help predict soil biological and ecological responses to multiple co-acting GCFs in addition to the number of factors; (ii) a larger dissimilarity among factors or larger number of factors will cause greater deviation of joint effects on soil properties and functions from expected effects; (iii) factor dissimilarity or number of factors may drive the emergence of factor interactions (synergistic or antagonistic).

## Results
### Effects of individual factors on soil functions and properties
The 12 single factors, our factor pool, produced a variety of responses on soil properties and functions, including positive, neutral and negative trends (Fig. 2a(1)–a(3) and Fig. 3a(1)–a(4)). However, none of the 12 single factors had significant effects on soil decomposition rate and four soil enzyme activity (Supplementary Data 1). Salinity and drought caused soil pH to increase ($P = 0.019$ and <0.001, respectively, Supplementary Data 1), while decreasing the proportion of water-

stable soil aggregates (WSA) ($P = 0.021$ and 0.042, respectively, Supplementary Data 1).

### Effects of multiple co-acting GCFs on soil functions and properties
The simultaneous effects of multiple factors on soil functions and properties changed directionally with an increase in the number of factors. When multiple factors were applied, WSA ($P < 0.001$ for all of 2, 5, and 8 factor groups) decreased (Fig. 2a(3), Supplementary Data 2), while soil pH ($P < 0.001$ for all of 2, 5 and 8 factor groups) increased compared to control (Fig. 2a(2), Supplementary Data 2). Soil decomposition rate decreased only in the eight-factor group ($P = < 0.001$) (Fig. 2a(1), Supplementary Data 2). Activity of β-glucosidase increased in all three factor groups ($P = 0.015$ for 2 factor group, $P < 0.001$ for 5 and 8 factor groups) (Fig. 3a(3), Supplementary Data 2). Phosphatase activity did not change in any factor group, while activity of N-acetyl-glucosaminidase and cellulase increased in 5, 8 factors groups and 8 factors group, respectively ($P = 0.017$, 0.002 and 0.008, respectively) (Fig. 3a(1) and a(2), Supplementary Data 2).

### Correlations of soil property and function responses to co-acting GCFs with factor dissimilarity
We used spearman correlation analyzes to show the changing trend of soil property and function responses along factor dissimilarity index range within each factor level. Factor dissimilarity was positively associated with soil pH, but negatively associated with WSA and soil decomposition rate (Fig. 2b(1)–d(3)). Soil enzymatic activity was positively correlated with the factor dissimilarity index (Fig. 3b(1)–d(4)). However, the correlation of soil responses with the factor dissimilarity index could be caused by the artefact that factor combinations with larger dissimilarity indices also have a higher chance of including the factors with extreme effect size. Therefore, the correlation analysis alone is insufficient to evaluate the true effect of factor dissimilarity.

### Hypothesis testing by hierarchical modeling framework
To test the possible drivers (number of factors and factor dissimilarity) of the variability of soil responses to simultaneously acting multiple factors and to separate the factor identity contribution, a hierarchical modeling framework was implemented based on machine learning and generalized linear model (GLM) algorithms (Supplementary Fig 2). To separate the contribution of factor identity, we first built the baseline model by predicting the soil functions and properties using the three null model predictions (i.e., predicted responses calculated by the responses of single factor treatments based on additive, multiplicative and dominative null model algorithms). Then, we tested the effect of factor dissimilarity and number of factors by adding additional predictors on the basis of the baseline model. The contribution of each predictor was evaluated by the increment of an R-squared value for Random Forest (RF) models and by comparing the changes of model AIC for the GLM.

The hierarchical modeling results based on the random forest algorithm showed that adding the number of factors improved the model $R^2$ for soil decomposition rate and three types of soil enzymatic activity (cellulase, β-glucosidase and phosphatase) (Fig. 2e(1), Fig. 3e(1)–e(4), Supplementary Data 3 and 4). Adding the dissimilarity indices further improved the model $R^2$ largely for soil decomposition rate and four types of soil enzymatic activity (Figs. 2e(1), 3e(1)–e(4), Supplementary Data 3 and 4). The permutation-based random forest approach also indicates that the importance of number of factors is significant for predicting soil decomposition rate and three types of soil enzymatic activity (cellulase, β-glucosidase and phosphatase), and factor dissimilarity index is significant for predicting all the soil responses except for WSA (Supplementary Data 5). The hierarchical modeling based on the GLM algorithm also showed similar results as

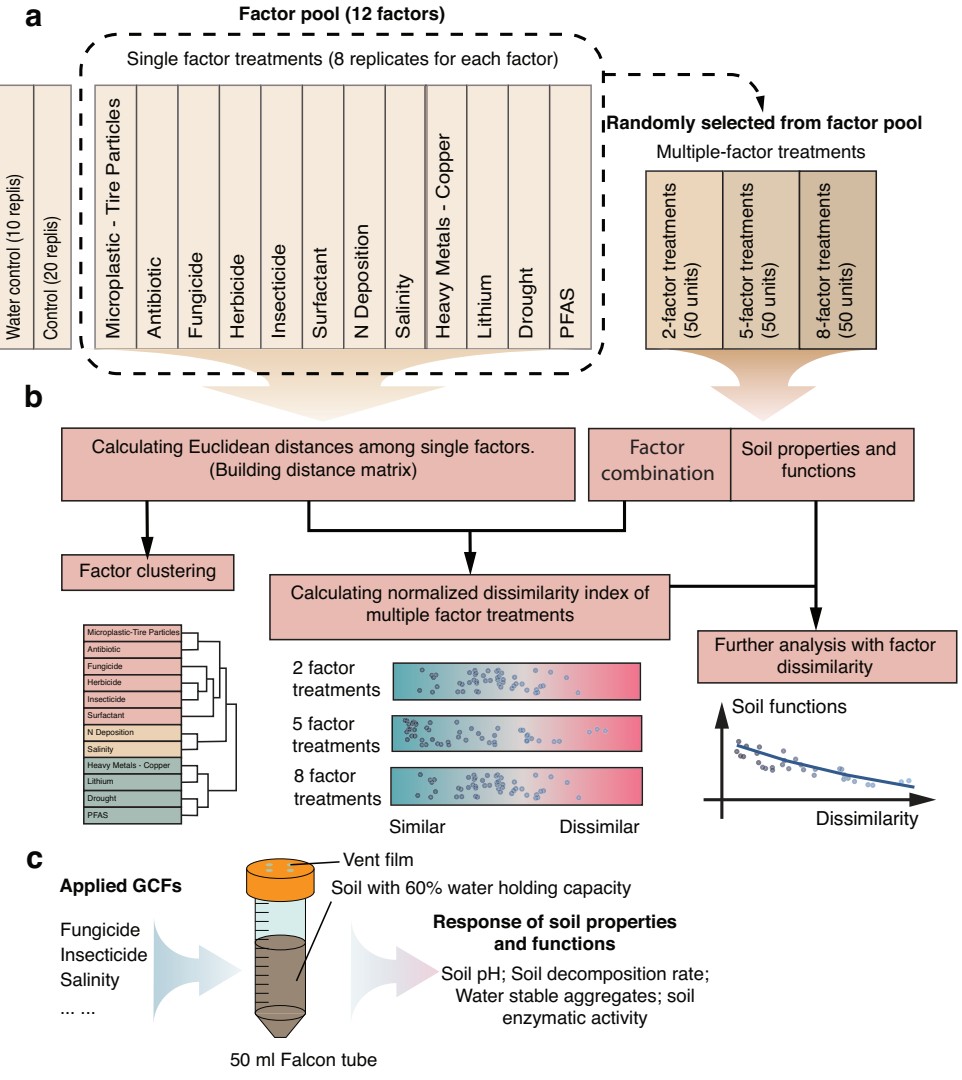

**Fig. 1 | Experimental design and analysis workflow. a** The design of the multiple factor experiment. There were 20 replicates for the control; 10 replicates for water control (without adding organic solvents); 8 replicates for single GCFs; 50 replicates for each factor level in the multiple factor group; the total number of experimental units = 20 + 10 + 8 × 12 + 3 × 50 = 276. Factor combinations in 2, 5, and 8 levels are randomly selected from the full combinations by drawing 2, 5, and 8 factors from the 12 factors pool without repetition. **b** Analysis workflow. The normalized dissimilarity index for each multi-factor treatment is calculated based on the Euclidean distances among single factors and the randomly selected factor combinations. **c** Experimental microcosm and response variables measured.

the RF models (Supplementary Data 7 and 8). Collectively, both machine learning and GLM algorithms indicate that the number of factors and factor dissimilarity are important predictors for the variability of soil responses to multiple GCFs.

**Emergence of factor net interactions in multiple-factor treatments**

We developed a methodology to assess the emergence of GCF interactions based on the deviation of soil responses from null model predictions (see methods and Fig. 4). Based on this approach, we identified the net interaction type of 150 multiple-factor treatments for each soil response. At the two-factor level, net interaction represents pairwise interaction. When the number of component factors is more than two, net interaction represents the overall effect of all pairwise interactions and higher-order interactions among factors. For soil decomposition rate, soil cellulase and β-glucosidase activity, based on three null model predictions, more synergistic net interactions emerged when the factor dissimilarity index increased (Fig. 5a(1)–a(3) and Fig. 6b(1)–b(3) and c(1)–c(3)). Across three number of factor levels,

no obvious change of the emergence of factor net interactions was observed (Supplementary Figs. 4 and 5).

To evaluate the drivers of factor interactions, we assessed the standardized deviations of soil responses from three null model predictions across the dissimilarity range and three factor levels, respectively. From the three null models, the model with the smallest sum of squared deviations (SSD) is used for estimating the deviation for each soil response. For soil decomposition rate, the additive model has the smallest SSD, while the dominative models have the smallest SSD for soil pH and WSA. For soil enzymatic activity, the multiplicative models have the smallest SSD, except for cellulase activity (dominative model) (Supplementary Data 9). The standardized deviation of soil decomposition rate from additive model predictions is correlated with the factor dissimilarity index, with synergistic interactions becoming more frequent with higher dissimilarity (Fig. 5). The standardized deviation of cellulase and β-glucosidase activity from the null models with the smallest SSD also show correlations with factor dissimilarity index, with more frequent synergistic interactions appearing with higher dissimilarity (Fig. 6). The soil decomposition rate responses to eight-

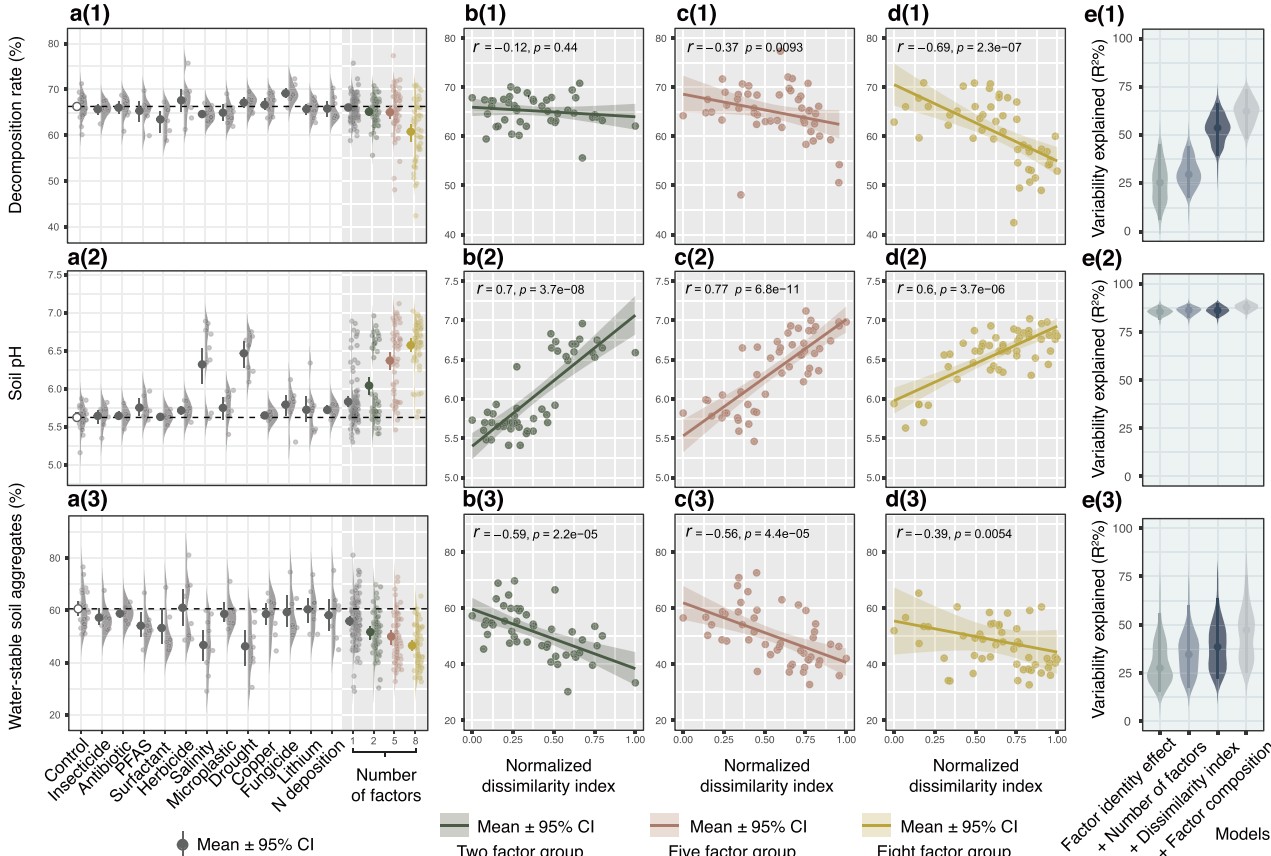

**Fig. 2 | Response of different soil properties to global change factors applied singly or in different numbers of factors (2, 5, and 8 co-acting factors).** For each soil property [**a**(1)–**e**(1), **a**(2)–**e**(2) and **a**(3)–**e**(3)], effect sizes of single factors ($n = 8$) and multiple factor groups (2, 5 and 8 factors, 50 treatments included in each factor group) were estimated [**a**(1)–(3)], then the correlations of soil property responses to the normalized factor dissimilarity index are shown in scatter plots [**b**(1)–**d**(1), **b**(2)–**d**(2) and **b**(3)–**d**(3)], Spearman correlation coefficients and significance of correlations are indicated by $r$ and $p$, respectively. The statistical test used was two-sided. [**e**(1)–(3)] show the soil response variability explained by Random Forest models ($R^2$). Added predictors represent the contributions of factor identity effect, number of factors, factor dissimilarity, and factor composition to the model explanatory rate. Points are the mean values and the error bars are their 95% confidence intervals (CI) (permutation $n = 1000$) (see Supplementary Data. 3).

factor treatments deviate significantly from the additive model (Supplementary Fig. 4), and also the soil enzymatic activity responses to higher number of factor treatments show significant deviation from the null models with the smallest SSD(Supplementary Fig. 5). These results suggest that the joint GCF effects on soil properties and functions deviate from null model predictions, and the number of factors and factor dissimilarity may drive the occurrence of more synergistic factor interactions.

## Discussion

By assessing soil ecological responses to a large set of factor combinations (150 different factor combinations) at three different factor levels (2, 5, and 8), our study suggests that, in addition to the number of factors, factor dissimilarity also drives the effects of multiple GCFs. Our study supports previous findings that the number of co-acting factors affects soil responses to GCFs[1,7]. As hypothesized, (i) the effects of factor dissimilarity played an important role in predicting the variability of soil responses to multiple GCFs; (ii) a larger dissimilarity among factors or larger number of factors will cause greater deviation of joint effects on soil properties and functions from null models; (iii) co-acting factors with higher dissimilarity tend to have more synergistic interactions. This provides a mechanistic perspective for predicting the joint effects of multiple GCFs on soil properties and functions and highlights the importance of systematically understanding the properties and mode of action of single GCFs. Our

findings also open opportunities towards improving management approaches; management should prioritize local GCFs not just in terms of the most severe factor(s), but should also take into account factor dissimilarity building on known factor traits (Supplementary Fig S8).

### Separating factor identity effects in multiple GCFs studies

A simultaneous manipulation of a large number of factors (usually more than six) is needed in multiple factor research to test general rules for multiple factor effects. However, the factor identity effect cannot be ignored, as it may drive contradictory results. The concept of "species identity effect" has been first raised in biodiversity studies for separating it from the diversity effect[17,18]. Similarly, in multiple factor studies, factors following the design we use here are randomly chosen from a factor pool. When the number of selected factors increases, there is also an increasing probability of including single factors with extremely strong effects in high-level factor combinations. This higher chance of including extreme factors in the higher number of factors group results in a "GCF-number effect" but this effect is not due to the number of factor effects, but an increased rate of selecting extreme single factors. When testing the effect of factor dissimilarity, similarly, because the factor dissimilarity index is calculated based on the single factor effects, a GCF with more extreme effect on soil properties and functions will also have larger effect 'distances' to other GCFs. Therefore, a factor combination that includes the extreme factor

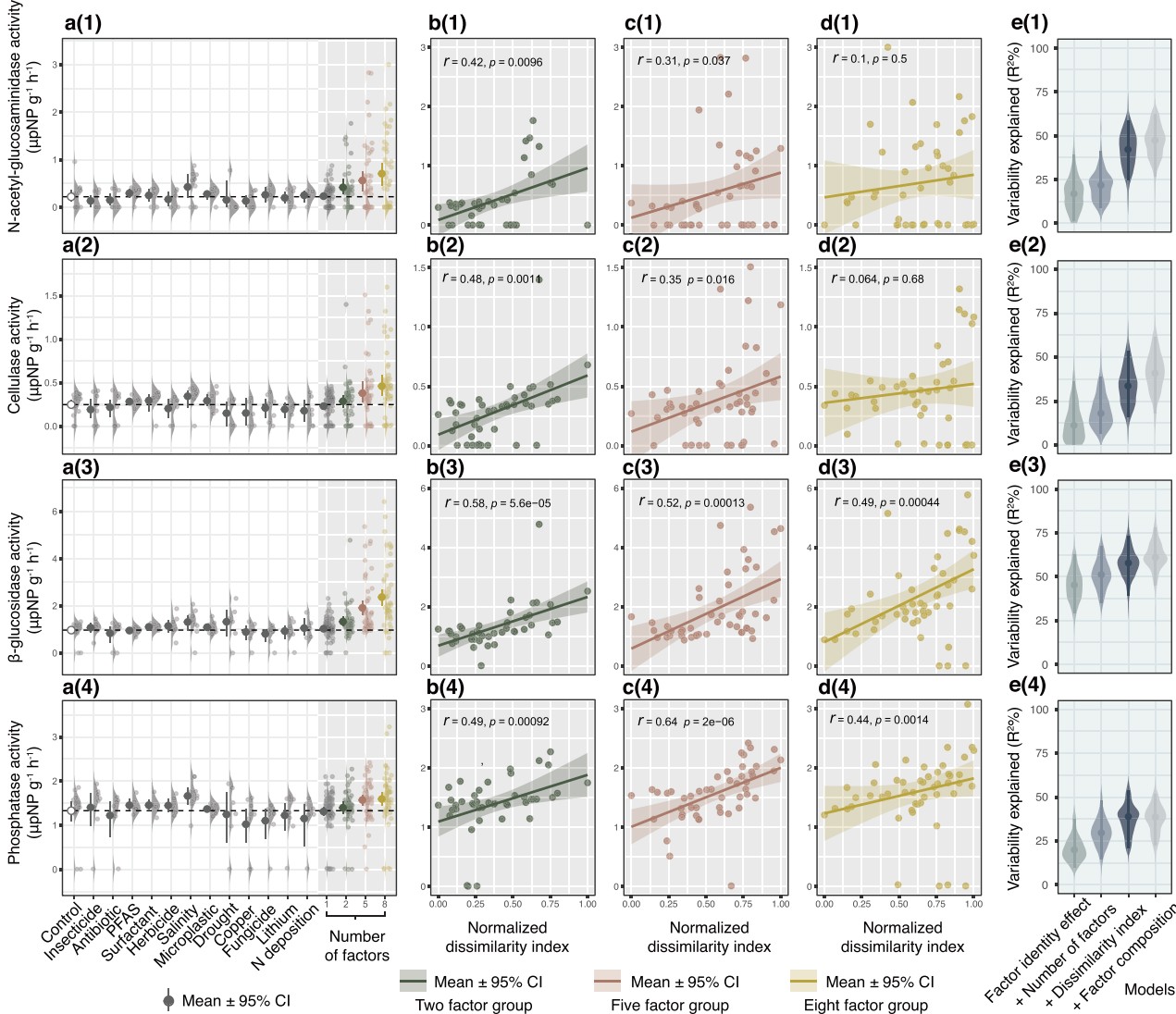

**Fig. 3 | Response of different soil enzyme activity to global changing factors applied singly or in different numbers of factors (2, 5, and 8 co-acting factors).** For each soil enzymatic activity [**a**(1)–**e**(1), **a**(2)–**e**(2) and **a**(3)–**e**(3)], effect sizes of single factors ($n = 8$) and multiple factor groups (2, 5 and 8 factors, 50 treatments included in each factor group) were estimated [**a**(1)–(3)], then the correlations of soil enzymatic activity responses to the normalized factor dissimilarity index are shown in scatter plots [**b**(1)–**d**(1), **b**(2)–**d**(2) and **b**(3)–**d**(3)], Spearman correlation coefficients and significance of correlations are indicated by $r$ and $p$, respectively. The statistical test used was two-sided. [**e**(1)–(3)] show the soil response variability explained by Random Forest models ($R^2$). Added predictors represent the contributions of factor identity effect, number of factors, factor dissimilarity, and factor composition to the model explanatory rate. Points are the mean values and the error bars are their 95% confidence intervals (CI) (permutation $n = 1000$) (see Supplementary Data. 4).

is likely to have stronger combined effect and at the same time has a relatively larger dissimilarity index. In this case, the correlations between factor dissimilarity indices and soil properties and functions could be only caused by the higher selecting rate of extreme factors, and are insufficient to support the effect of factor dissimilarity. Thus, an appropriate statistical method is needed for disentangling the effect of factor number and dissimilarity from the factor identity effects.

Although only a few studies on multiple GCFs have assessed factor identity effects[8,9], attempts have been made in some studies to use both parametric and nonparametric methods[1,6]. Inspired by identifying contributions of species identities, previous work has employed a hierarchical diversity-interaction modeling framework based on linear mixed-effects models to assess the contribution of GCF identities by ANOVA tests[6]. But the limitation of traditional statistical models is that they lack the power to capture unknown nonlinear patterns of the hypothesis in higher dimensionality[19]. Another way to identify the

individual factor contribution is comparing observed effects with effects predicted by null models. Null models assume that there are no interactions among factors[20], thus the predictions of null models can be viewed as the combinations of single factor effects. To better deal with potential nonlinear relationships, a previous study explored the potential to combine machine learning algorithms with null model predictions to address the effects of factor number and to differentiate the factor identity contribution[1]. In our study, since null model prediction results have been integrated as the baseline models in the hierarchical modeling framework, the increase of the model predictability from baseline models can be interpreted as the contribution of factor number effect or factor dissimilarity effect other than the factor identity effect. Additionally, we further analyzed the change of standardized deviation of factor joint effect from the best-fitting null model across the dissimilarity index range, and indicated the direction of deviations (the types of the emergent interactions). In this way, the effect of factor dissimilarity can be depicted by both the explained

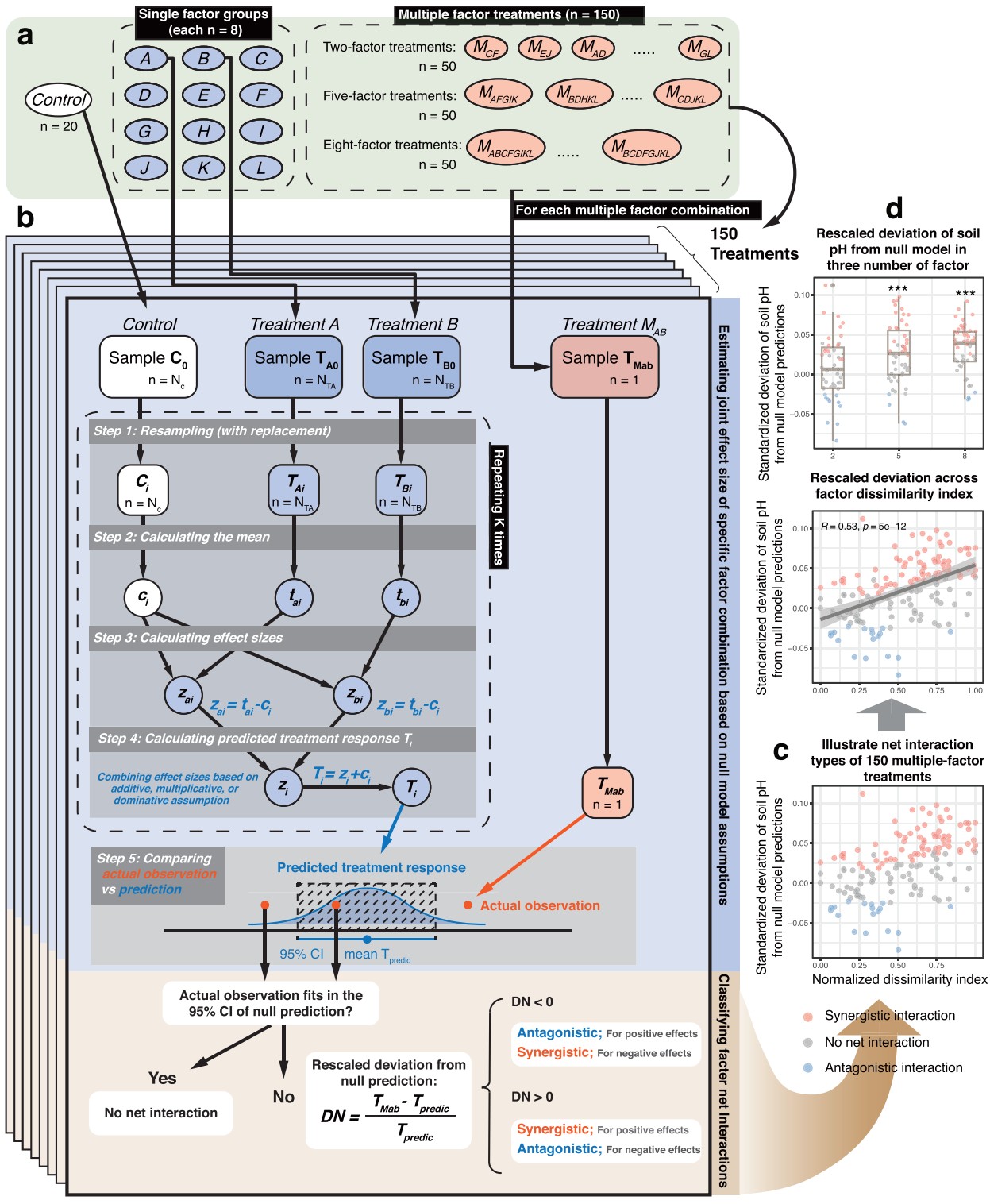

variability increased from the baseline model and the changes of standardized deviation of soil responses from null models, but not directly from the correlation of factor dissimilarity with soil responses. For example, in our study, soil pH is strongly associated with factor dissimilarity, but the correlation is mostly caused by the contribution of extreme factors, because the baseline model can explain 85.4% of the response variability and adding other predictors hardly increase model R squared (Fig. 2b(2)−e(2) and Supplementary Data 7). WSA is also negatively correlated with the dissimilarity index. However, from the standardized deviation analysis of WSA from the dominative null

model prediction, we find the WSA responses are mostly subjected to the dominative model and there is no correlation between factor dissimilarity and standardized deviation from the dominative model (Fig. 5c(3) and Supplementary Data 9). Because the factor identity effect caused contradictory results, we suggest separating factor identity effects from other effects driven by multiple GCFs. Our study also provides a practical method to evaluate the contribution of factor identity effects and it should be used as the baseline for testing other hypotheses in future multiple GCFs studies based on randomly-drawn factors.

**Fig. 4 | Calculating soil response deviation from null model prediction and net interaction type classification for 150 multi-factor treatments. a** Treatments in the experimental design. Single factor treatments are shown in blue ovals. Each multi-factor treatment is shown by a red oval. The subscript of a multi-factor treatment indicates the component factors. **b** Interaction type classification workflow for multi-factor treatments. The workflow includes two parts: (1) estimating the joint response distributions of component factors of multi-factor treatments; (2) identifying the net interaction type for multi-factor treatments. For illustration purposes, one two-factor treatment (includes factor A and B) is taken as an example. In Step 1, we resampled from each control, single factor A and B treatment with replacement to generate $C_i$, $T_{Ai}$ and $T_{Bi}$. Then, in Step 2, mean values of each resampled treatment ($c_i$, $t_{ai}$ and $t_{bi}$) are calculated. In Step 3, absolute effect sizes from control ($Z_{ai}$ and $Z_{bi}$) for A and B single factor treatments are calculated. In Step 4, combined effect size of A and B ($Z_i$) are calculated depending on different null model assumptions (additive, multiplicative or dominative). Then the control mean is added to $Z_i$ to generate predicted joint response

($T_i$). Steps 1-4 are repeated $K$ times to generate the distribution of the predicted joint response of factor A and B. Then in Step 5, we compared the actual joint response of factor A and B ($T_{Mab}$) to the predicted response distribution. If the actual observation fitted within the 95% confidence intervals (CIs) of prediction distribution, then it was regarded as no net interaction. If it did not fit, then we calculated the rescaled Deviation from Null model prediction (DN). Then we classified the net interaction type based on the rescaled DN. **c** Visualization of the rescaled DN and net interaction types of 150 multi-factor treatments. **d** Statistical analysis of rescaled DN of soil response across factor dissimilarity index and in three different number of factor groups. Two-sided t-tests were performed for three factor groups. Asterisks represent the statistically significant difference of treatment group from zero ($n = 50$): ***$P < 0.001$, **$P < 0.01$, *$P < 0.05$. Boxplots showing the distribution of deviations across factor groups. The box spans the interquartile range (IQR) with the median indicated by the line inside. Whiskers extend to the minimum and maximum within 1.5 times the IQR. Outliers are shown as dots.

## The role of factor dissimilarity in driving the emergence of GCF interactive effects on soil properties and functions

We found increased emergence of synergistic factor interactions on soil decomposition rate, soil cellulase and β-glucosidase enzymatic activity when factor dissimilarity increases. By analyzing the deviation of factor joint effects from the best-fitting null model predictions across the dissimilarity index range, we found that factor dissimilarity drives the interactions of multiple GCFs towards a more synergistic direction (Figs. 5a(1), 6b(3), c(2)). Our finding indicates that factor dissimilarity underpins the interactive mechanisms of GCFs on soil properties and functions.

The role of factor dissimilarity in driving the interactive effects of multiple GCFs could be due to three distinct mechanisms. Firstly, factors that differ in their physicochemical nature may be more likely to have direct interactions compared to factors with the same physicochemical nature. These direct factor interactions are only related to the physical or chemical properties of the factor itself, without considering how they affect soil organisms and processes[21]. For example, drought can interact with other chemical factors as a concentration amplifier[22], and surfactants can increase the solubility or movement of organic pollutants[23]. Direct factor interactions usually amplify the intensity of single factors and, thus, the emergence of synergistic effects of high-dissimilarity multiple factors may be derived from the occurrences of direct factor interactions.

Secondly, factor dissimilarity may play a role in affecting species adaptation to a multiple-GCF environment. Performance trade-offs are common when populations are exposed to multiple-factor environments[24,25]. Based on the pareto optimality theory, species cannot optimize adaptation to multiple factors at the same time[26,27]. When factors are more dissimilar, these trade-offs would be larger, leading to a lower overall adaptation performance to the multiple-factor environments. By contrast, when the effects of factors are similar, the adaptation strategy of a population to one factor could also allow its adaptation to another factor by applying the same genetic or metabolic responses (e.g., cross-protection)[28,29].

Thirdly, factor dissimilarity could reshape the co-tolerance space of species to multiple factors. In the co-tolerance theory framework, the resistance of a community to multiple-stressor environments is affected by the relatedness of species' tolerances to different stressors[30]. If species tolerances to stressors are positively correlated, the overall species loss will be less than if the tolerances are unrelated, but if species tolerances to stressors are negatively correlated, more species (and the functions they drive) will be lost in multi-stressors environments. When multiple factors are quite different, their mode of action or target species range might also be different. From the perspective of the community, species tolerances to factors would be more negatively correlated. In this scenario, more species from the community are likely to be lost when more dissimilar GCFs are applied

simultaneously. Biodiversity loss likely leads to reduced ecosystem functions (e.g., litter decomposition rate) based on biodiversity insurance theory[31]; this may explain why there are more synergistically negative effects on soil decomposition rate when factor dissimilarity indices are higher (Figs. 5 and 6).

Our study investigated the effects of factor dissimilarity on soil properties and ecological functions, suggesting that factor dissimilarity can drive more frequent occurrence of synergistic effects on several soil functions. Future work should address the effects of factor dissimilarity at different levels of the ecological hierarchy (organism, population and community levels), or the temporal variation in effects of factor dissimilarity. Incorporating the effect of factor dissimilarity in future multiple GCFs studies will be helpful for estimating effects of co-acting GCFs and may be useful in informing protocols for ecosystem management and restoration.

## Methods
### Experimental design
The experiment was set up with a GCF pool that includes 12 factors: salinity, drought, microplastic, fungicide, herbicide, antibiotic, insecticide, surfactant, nitrogen deposition, heavy metal pollution, perfluoroalkyl and polyfluoroalkyl substances (PFAS), lithium. The selected factors were chosen from the most frequently occurring anthropogenic factors in soil ecosystems subject to intense human influence[32,33], and differ in intrinsic features (physical, chemical etc.) and effect mechanisms (mode of action, effect targets etc.) in affecting soil properties and functions[10,11]. Detailed information about the selected GCFs is presented in the supplementary information. On the basis of previous experimental designs[1,34,35], three multi-factor levels (2, 5 and 8 factors, 50 replicates) were created by a random factor-selection method. To achieve this, first, complete sets of factor combinations for each factor level were generated (e.g., for the 5 factor level, there were in total 792 different factor combinations for choosing 5 factors from a 12 factor pool). Then we randomly selected 50 factor combinations from all the possible combinations at each factor level without replacement to avoid selecting repeated factor combinations. Furthermore, we set up single factor treatments with 8 replicates for each factor and the control group including 20 replicates. Finally, to test for the effects of organic solvents (dimethyl sulfoxide (DMSO) and acetone) used to apply chemical GCFs (fungicide and herbicide) on soil properties and functions, we included 10 additional replicates (water control) that received the same rate of water instead of organic solvent in the experiment. Collectively, we had $(50 \times 3) + (12 \times 8) + 20 + 10 = 276$ units in our experiment (Fig. 1a).

### Soil preparation and incubation system
The soil used in the experiment was collected in February 2022 from a local grassland at an experimental site of Freie Universität Berlin

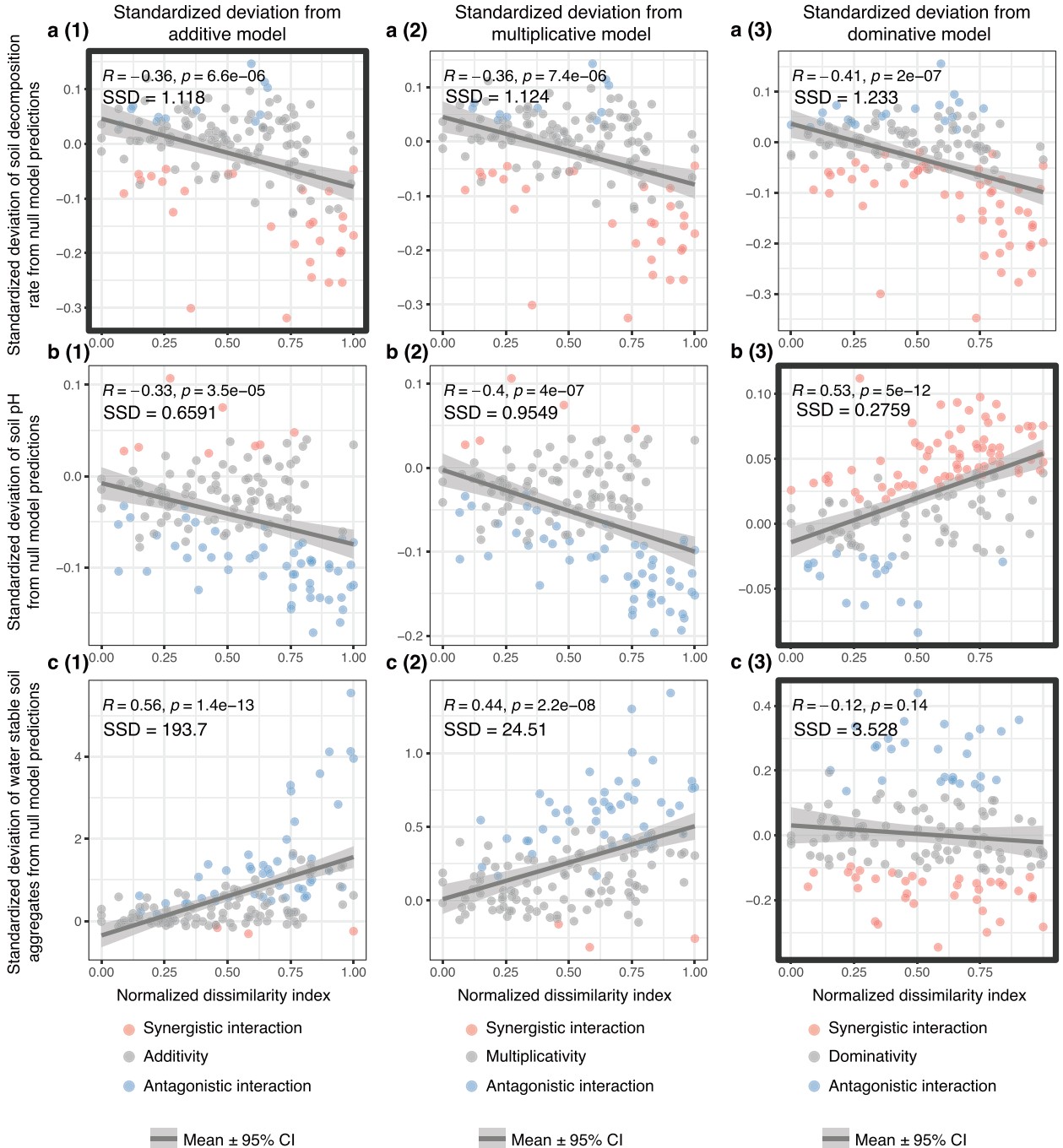

**Fig. 5 | Correlation between rescaled deviation of soil responses (decomposition rate, soil pH and WSA) from three null model predictions and normalized dissimilarity index. a**(1)−**c**(3) Scatter plots show the standardized deviation of soil responses from null model predictions for multiple-factor treatments. Net interaction type of each multiple-factor treatment is marked as different colored points (antagonistic, blue; synergistic, red; no interaction, gray). The better-fitting null model for each soil response has been selected based on the smallest model sum of squared deviation (SSD), and it is indicated by the bold frame [**a**(1), **b**(3) and **c**(3)]. The correlations between standardized deviation of soil responses from null model predictions and normalized dissimilarity index are shown in linear correlation with 95% confidence intervals (CIs). The statistical test used was two-sided. Spearman correlation coefficients and significance of correlations are indicated by R and *p*, respectively.

(52° 28' N, 13° 18' E, Berlin, Germany) with a sandy loamy texture. A sandy loamy texture refers to soil that contains a balanced proportion of sand, silt, and clay particles, with sand being the predominant component. Before the start of the experiment, the soil was air dried and passed through a 2 mm sieve to remove large stones and big grass roots. To prepare the "loading soil" for the factor implementation, one eighth of the air-dried sieved soil was sterilized at 121 °C for 20 min. Loading soil was used to more

effectively mix small amounts of chemicals into the experimental units; it was sterilized to avoid large local effects of concentrated chemicals on soil microbes.

The experimental unit was a 50 mL mini-bioreactor (Product Nr: 431720, Corning®, USA) with a vented film, which allows gas exchange but prevents microbial contamination (Fig. 1c). Inside the bioreactor we added 40.0 g (dry weight, d.w.) soil with the respective GCF treatments.

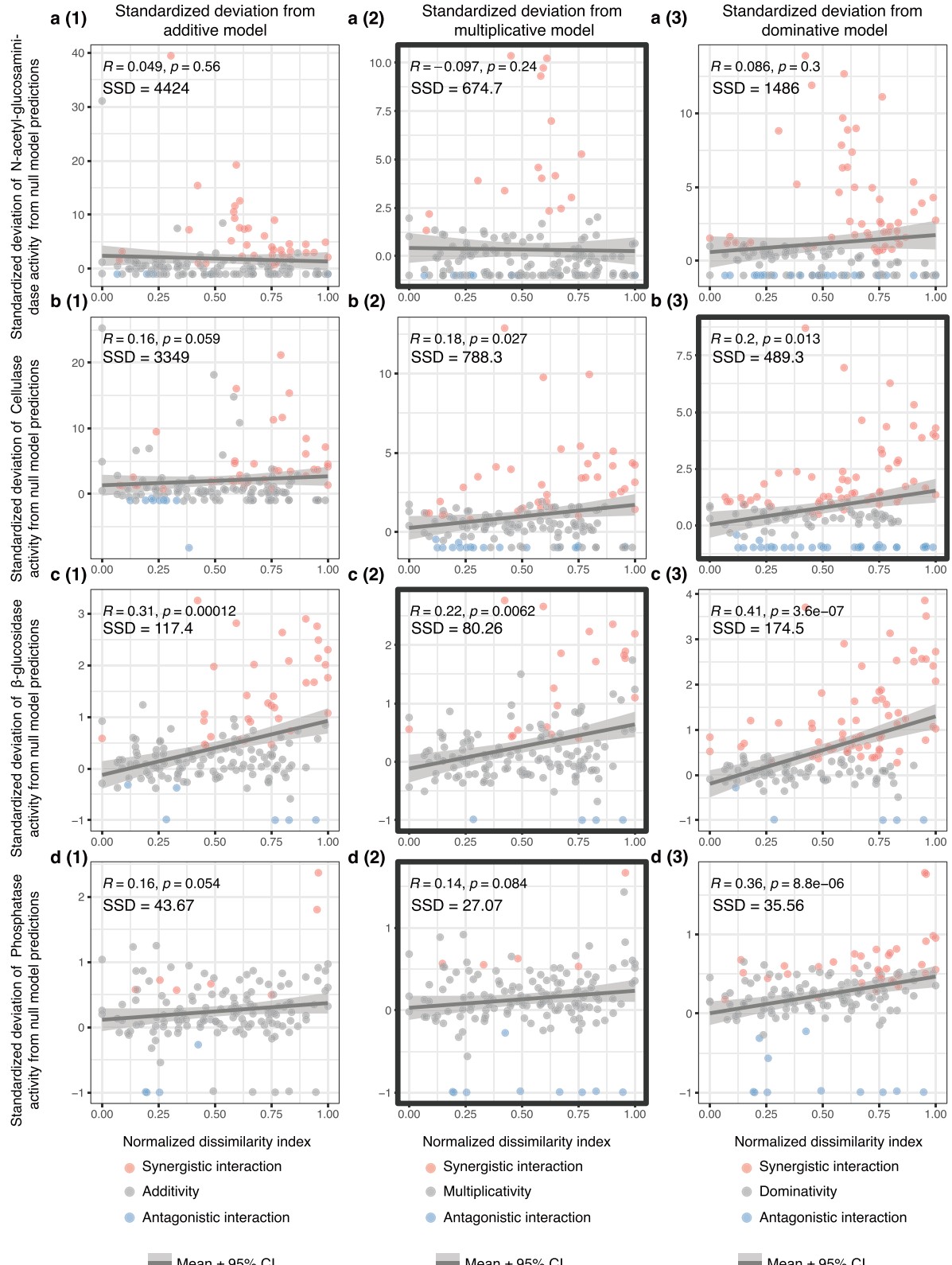

**Fig. 6 | Correlation between rescaled deviation of soil enzymatic activity from three null model predictions and normalized dissimilarity index. a**(1)−**d**(3) Scatter plots show the standardized deviation of soil enzymatic activity from null model predictions for multiple-factor treatments. Net interaction type of each multiple-factor treatment is marked as different colored points (antagonistic, blue; synergistic, red; no interaction, gray). The better-fitting null model for each soil enzymatic activity has been selected based on the smallest model sum of squared deviation (SSD), and is indicated by the bold frame [**a**(2), **b**(3), **c**(2) and **d**(2)]. The correlations between standardized deviation of soil enzymatic activity from null model predictions and normalized dissimilarity index are shown in linear correlation with 95% confidence intervals (CIs). The statistical test used was two-sided. Spearman correlation coefficients and significance of correlations are indicated by R and p, respectively.

## Implementation of GCFs and harvest

We included 12 GCFs in the factor pool: salinity, drought, microplastic, fungicide, herbicide, antibiotics, insecticide, surfactant, nitrogen deposition, heavy metals, PFAS and lithium. We used 30% water holding capacity to represent the drought treatment, while other units kept 60% water holding capacity for the entire incubation period[1]. To simulate a copper contamination spot, we added copper using Copper (II)- sulfate- pentahydrate (CAS: 7758-99-8) in distilled water to reach a final concentration of 100 mg Cu/ kg dry soil[1]. For microplastic treatment, we picked tire wear particles as the microplastic pollution with a concentration of 0.1% (w%w)[36]. To simulate the annual N accumulation rate of 100 kg N ha$^{-1}$ yr$^{-1}$, we assumed that the surface 10 cm depth soil acted as the recipient of all N deposition. We used a one-time addition method to add dissolved form of ammonium nitrate (>98%, p.A., ACS. Roth GmbH, Karlsruhe, D. article K299.1) to reach the final ammonium nitrate concentration of 439.6 mg kg$^{-1}$ in soil. For soil salinization, we chose the 4.0 dS/m conductivity to represent a soil salinization situation with the approach of mixing solid NaCl with the soil[37]. For herbicide treatment, we took diflufenican, a typical commercial herbicide, with the application rate of 1.0 mg/kg and dissolved it in acetone to simulate the field application[38]. A field application rate of Carbendazim (PESTANAL ® analytical standard, Sigma-Aldrich, MO, USA, catalog #45368) at 6.0 mg/kg was selected as the application fungicide dosage in our experiment[39]. We adopted the concentration of antibiotic from a previous experiment as 3.0 mg oxytetracycline (Sigma-Aldrich, MO, USA, catalog #PHR1537) per kg soil[1]. We took a moderate dosage of 50 ng g$^{-1}$ dry soil imidacloprid (PESTANAL ® analytical standard, Sigma-Aldrich, MO, USA, catalog #37894), one of the most common neonicotinoid pesticides, to simulate the pollution of pesticides[1]. Perfluorooctanoic acid (PFOA) concentration in surface-soil was reported up to 50 mg/kg[40]. Around 1 ppm perfluorooctanoic acid (CAS: 335-67-1) was used to simulate the existence of PFOA in agricultural soil in Germany[41,42]. In the context of sandy soil, the toxicity of surfactant occurred at a concentration of 16 mg kg$^{-1}$ in a field experiment[43]. Therefore, we used 16 mg kg$^{-1}$ sodium dodecylbenzenesulfonate (CAS: 25155-30-0) to simulate a surfactant contaminated hotspot in agricultural soil. In European agricultural soil, it has been reported that the mean concentration of Li is approximately 11.4 mg/kg[44,45]. We applied 11.4 mg kg$^{-1}$ lithium chloride in solution form to the soil to simulate a Lithium contaminated soil.

To homogeneously mix GCFs with the testing soil, we added chemical factors to 5.0 g (d.w.) loading soil first then mixed the loading soil with the other 35.0 g (d.w.) soil. Concentrated solutions were prepared for every chemical factor except for salinity and microplastic. Most of the chemicals we used were dissolved in distilled water, except for fungicide (carbendazim, dissolved in DMSO) and herbicide (Diflufenican, dissolved in acetone). According to the designed factor combination for each treatment, 100 μL solution (water, DMSO or acetone) carrying appropriate chemical dose for 40.0 g (d.w.) soil was added to 5.0 g (d.w.) loading soil inside a 150 mL cup. To standardize the amount of solvents we added into every treatment, treatments which had fewer factors, for instance, single factor treatments and control treatments, additionally received solvents (water, DMSO or acetone) to the same amount of solvents added in the eight factor treatments. To further test the effects of the solvents (DMSO and acetone) on soil properties and functions, another 10 control treatments only received the same amount of distilled water. The effects of organic solvents on soil properties and functions are shown in Supplementary Fig. 6. For the experimental units that include microplastic or salinity treatments, 40.0 mg tire particles (1–2 mm diameter) or 200.0 mg sodium chloride were added into the 150 mL cup accordingly. Then an additional 35.0 g of air-dried soil was added to every 150 mL cup. After covering with a cap, all the soil treatments were mixed for 30 min with a shaking machine (Product Nr: 541-21009-00,

Reax2, Heidolph Instrument GmbH & Co. KG, Schwabach, Germany) at a speed of 80 rpm to achieve a homogeneous distribution. After the mixing process, the soil-chemical mixture was transferred to the 50 mL mini bioreactors. For tracking the litter decomposition rate during the experiment, a sterilized tea bag was placed vertically in the center of the soil. Finally, distilled water was added to bring the soil water content to 60% of the soil water-holding capacity (30% of water-holding capacity for drought treatments).

All 50 mL mini-bioreactors were incubated at 25 °C in a dark environment for 42 days. As there was on average 0.5 g weight loss every week for each mini-bioreactor, we added 0.5 mL distilled water to each treatment every week to keep the water content constant. After 42 days, all units were harvested. Soil cores were taken from the bioreactors, the tea bags (see below) were removed and the soil of each treatment was homogeneously mixed by a spoon in a sterilized Petri dish for 2 min. 5.0 g fresh soil was collected and stored at 4 °C for enzymatic activity measurement, and the remaining soil was air dried at room temperature for soil property measurements. Tea bags from all units were collected and oven dried (60 °C) before measuring litter decomposition rate.

## Soil response variables

The soil response variables we measured in this experiment are: litter decomposition rate, soil pH, WSA, and the activity of four extracellular soil enzymes. The litter decomposition rate was measured and calculated using the tea bag index method[46]. Briefly, a sealed tiny bag (with 38 μm mesh size) containing 300.0 mg (d.w.) tea biomass was placed into the soil, then the proportional weight loss was calculated based on the tea biomass (d.w.) inside the bag before and after the incubation. For the soil pH, 5.0 g air dried soil was mixed with 25 mL distilled water within a 50 mL centrifuge tube, and then the pH of the soil suspension was measured by a pH meter (Hanna Instrument, Smithfield, USA). The proportion of WSA was measured following a modified protocol[47]. The measurements of N-acetyl-glucosaminidase (chitin degradation), cellulase (cellulose degradation), β-glucosidase (cellulose degradation) and phosphatase (organic phosphorus mineralization) activity followed a high throughput microplate protocol[48] using a microplate reader (BioRad, Benchmark Plus, Japan).

## Effect size calculation and significance test of single and multiple factor groups

Data were analyzed with R Version 4.1.1[49]. For single factor and multiple factor groups, the effect size and 95% confidence intervals (CIs) of each group were estimated with a nonparametric bootstrap method with 10,000 permutations[50]. Considering the multiple testing problem, the statistical significance of single and multiple factor effects was evaluated by using adjusted P-values based on the Benjamini-Hochberg method[51].

## Calculating factor dissimilarity

We used the "vegan"[52] R package to calculate the Euclidean distances between all pairwise factor combinations based on the corresponding standardized effect sizes of singly applied factors on the seven soil properties (including four soil enzyme activity, WSA, soil decomposition rate and soil pH). Clustering of single factors was conducted based on Euclidean distance by using hierarchical clustering analysis ("ggdendro"[53] and "dendextend" R packages were used)(Supplementary Fig. 1a). Then, we used principal coordinate analysis to visualize the distances among factors, resulting in the PCoA1 and PCoA2 axes explaining 55.43% and 23.38% of the variation respectively (Supplementary Fig. 1b).

For the multiple-factor treatments, we calculated a dissimilarity index (DI) for each unique factor combination by adding up the Euclidean distances between every two component factors in the

multiple-factor treatments,

$$DI_i = \sum_{j \in N_i} d_j \tag{1}$$

where $DI_i$ ($i = 1, 2, \ldots, 50$) is the dissimilarity index of multiple-factor treatment's $i$-th combination in a specific level of number of factors (2, 5, and 8), $d_j$ is the Euclidean distance between the $j$-th two factor pair estimated based on the single factor experiment, and $N_i$ is the set of all unique factor pairs of the treatment $i$.

To compare dissimilarity indices between different numbers of factor levels, we normalized the dissimilarity indices of each factor level to a range between 0 and 1 by using the "range" method of the preProcess function from the "caret" R package[54]. To do this, we subtract the minimum value from each dissimilarity index and divide it by the range of the dissimilarity indices of each number of factor level. The distributions of normalized factor dissimilarity indices in three factor levels and the reasons for choosing the normalization method are shown in Supplementary Fig 7.

### Correlations between soil responses and factor dissimilarity within factor levels

To show the changing trend of soil property and function responses across the range of factor dissimilarity within factor levels, we applied Spearman correlation analyses to the normalized dissimilarity index and soil properties and functions in each factor level. Estimated $P$ value and coefficient are provided respectively for each correlation.

### Predicting effects of multiple co-acting factors by null models

In ecological studies, null models are used for predicting the joint effect of multiple factors without considering interactions[55]. For commonly-used null models, the additive model assumes that the joint effect of multiple factors will be the sum of the effects of the single factors, indicating that the sensitivities of the target to factors are negatively correlated. The multiplicative model assumes that the effects of single factors are combined by proportional change, meaning that the factor sensitivities are non-correlated. In the dominative model, the factor with the largest absolute effect overrides other factors, implying the factor sensitivities are positively correlated. To make plausible predictions of multiple-factor effects on soil responses, we imposed three null model assumptions (i.e., additive, multiplicative, and dominative assumption) for generating predictions for the multiple factor treatments instead of arbitrarily selecting one[20]. For each null model assumption, we applied the calculation methods from a previous study[1]. For each number of factors level, the unique subset of factor combinations randomly chosen from the 12 factor pool is denoted as $A_n$ ($n = 2, 5, 8$). For each multiple-factor combination $K_m \in A_n$ (e.g., $K_1 = $ [Microplastic, Drought], $K_2 = $ [Antibiotic, Fungicide] ..., $K_{50} = $ [Salinity, PFAS]; $K_1, K_2, \ldots K_{50} \in A_2$), $K_m$ includes $N$ component factors, denoted as $(F_{m_1}, F_{m_2}, \ldots F_{m_N})$ ($N = 2$ for $A_2$, $N = 5$ for $A_5$, $N = 8$ for $A_8$). $ES_{m_i}$ is the mean of estimated effect size of the factor $F_{m_i}$ observed from the single factor treatment. In additive assumption, predicted effect size of factor combination $K_m$,

$$P_{additive_m} = \sum_{i=1}^{N} ES_{m_i} \tag{2}$$

Considering each set of $A_n$ has 50 elements ($K_m$), we applied a bootstrapping method (with 1000 iterations; see Fig. 4) for each $K_m$. Each $K_m$ has 1000 iterated effect size predictions, in total 50,000 effect size predictions were made for all treatments for each number of factors level, which should be sufficient for generating reliable estimates. Afterwards, the mean value and 95%CI were calculated from the distribution of each factor combination. The same bootstrapping procedures were used in multiplicative and dominative assumptions.

For multiplicative assumption, based on a previous method[20], the predicted effect size of factor combination $K_m$ is shown as:

$$P_{multiplicative_m} = CT \prod_{i=1}^{N} \left(1 + \frac{ES_{m_i}}{CT}\right) - CT \tag{3}$$

$CT$ is the estimated response of the control group. For the dominative null models, the predicted effect size is

$$P_{dominative_m} = ES_{m_i} \ (|ES_{m_i}| = \max(|ES_{m_1}|, |ES_{m_2}|, \ldots |ES_{m_n}|)) \tag{4}$$

### Hierarchical modeling framework for hypothesis testing

To disentangle the contribution of possible drivers (number of factor effect and factor dissimilarity effect) on the variability of soil properties and functions in response to multiple GCFs, a hierarchical modeling framework was implemented (Supplementary Fig 2). To generate robust results, we compared the modeling results generated by both machine learning[56] and GLM. In the modeling, data from all the treatments except for the controls were used. In both algorithms, to separate the factor identity effects, the null model predictions (from additive, multiplicative, and dominative models) were first included as predictors in the baseline model (Model 1), which is regarded as the soil response variability explained by the contributions of factor identity. Number of factors was solely included as the predictor in Model 2, and in Model 3 factor dissimilarity indices were included instead. Then, on the basis of the baseline model, each soil response was modeled by adding the number of factors or factor dissimilarity indices as an additional predictor in Model 4 and Model 5 respectively. Furthermore, in Model 6, both factor dissimilarity indices and number of factors were added on the basis of the baseline model. Lastly, factor composition (i.e., a binary matrix coding the features for each treatment, where 1 or 0 represent the presence or absence of each stressor.) was included as the last predictor for the final model (Model 7). The formula describing each model is shown in Supplementary Data 6. In Model 7, due to different model algorithms, factor composition has a different meaning. For the Random Forest algorithm, the factor composition stands for all the information from the experimental design (also includes the information of other predictors, e.g., number of factors), and theoretically it can provide the best model fits. Thus, in the hierarchical modeling framework, the factor composition is only being added at the end to show the variability that can be explained by the experimental treatments (the randomly-drawn factors). In the GLM, including factor composition does not stand for the factor identity effects and also does not have a specific statistical meaning in this case. But for comparison to the Random Forest model, we still provide the modeling results of Model 7.

We evaluated the variability of soil responses explained by all seven models with model R-squared values ($R^2$, %). To evaluate the contribution of each model predictor, for the GLM, we compared models by their AIC (Akaike information criterion) values based on the ANOVA tests (Supplementary Data 7 and 8) and evaluated the increase in model R-squared values. For the Random Forest models, the contribution of each model predictor was evaluated by the increase in the model R-squared values. To address the statistical inference of predictor contributions in the Random Forest models, we used a permutation-based random forest model approach with 1000 permutations to calculate the relative importance of each predictor[19,57]. Adjusted $p$ values of relative importance for each model predictor are shown in Supplementary Data 5.

### Identify factor net interactions and quantify deviations of joint effects from null model predictions

To understand the joint effects of multiple factors, interactions among multiple factors can be evaluated by the deviation of experimental

observation of soil responses from null model predictions[58]. For each soil property or function, we identified the net interaction type of all 150 tested factor combinations. The net interaction represents the overall effect of all interactions among component factors, including all pairwise interactions and high-order interactions. For classifying the type of the net interaction, we applied a framework for measuring ecological stressor interaction[58] and modified it for our research objectives (Fig. 4). In our study, we first compared the observed soil responses with the 95% CIs of null model predictions (generated by bootstrapping method with 1000 permutation) (Fig. 4 b). Observations that fit into the 95% of the null model CIs were classified as no interaction, others were categorized as antagonistic or synergistic net interaction depending on the direction of the deviation. For variables with positive responses to multiple GCFs (including four soil enzyme activity, soil pH), observations that did not fit in null model predictions were classified as synergistic interactions when rescaled Deviation from Null model (DN) > 0, and as antagonistic interactions when DN < 0. For negative responses (including soil decomposition rate and WSA), they were classified as synergistic interactions when DN < 0, but as antagonistic when DN > 0. Additionally, because there is still insufficient knowledge of choosing suitable null model assumptions for higher levels of ecological organization targets (e.g., community or ecosystem functions)[59], we assessed the overall deviation of all multiple-factor treatments from three null model assumptions (additive, multiplicative and dominative), respectively. The null model that had the smallest sum of the squared deviation from the treatment responses was selected as the best-fitting null model for a certain soil response (Supplementary Data 9). To test the effects of the number of factors and factor dissimilarity on driving factor interactive effects, statistical analysis was implemented for standardized deviations across the number of factor levels (Supplementary Data 10) and the factor dissimilarity range (Figs. 5 and 6).

## Reporting summary

Further information on research design is available in the Nature Portfolio Reporting Summary linked to this article.

## Data availability

The data that support the findings of this study are available in the Github repository, with the identifier https://doi.org/10.5281/zenodo.13384438. Source data are provided with this paper.

## Code availability

Documented code and data to replicate all analyses in this manuscript is available from a released GitHub repository: https://doi.org/10.5281/zenodo.13384438.

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

## Acknowledgements

M.C.R. acknowledges support from an European Research Council Advanced Grant (grant number 694368) and from the European Union projects MINAGRIS and PAPILLIONS, as well as from the BMBF-funded project µPlastic. M.B., H.L., and Y.Z. acknowledge the China Scholarship Council for a scholarship (202106050016, 202108080156 and 202106320023, respectively).

## Author contributions

M.H.B.: design of the study, experiment setup, analysis of data, and writing. H.L.: design of the study, experiment setup, analysis of data, and writing. P.M., and Y.Z.: design of the study, experiment setup. M.R.: review and editing. M.C.R.: conceptualization, review, and editing.

## Funding

## Competing interests

The authors declare no competing interests.
