## [Peer Review File · Nature Communications]

Number and dissimilarity of global change factors influences soil properties and functionsEditorial Note: This manuscript has been previously reviewed at another journal that is not operating a transparent peer review scheme. This document only contains reviewer comments and rebuttal letters for versions considered at Nature Communications.

Reviewers' Comments:

Reviewer #1:

Remarks to the Author:

General

- I am mostly pleased with the revisions, and I am also happy to see that the authors have removed the groupings of factor dissimilarity in favour of a continuous gradient.
- My main issues currently are to do with the language and presentation/flow of the manuscript. The authors with stronger English skills need to take another look over the text, as many parts are awkwardly-written and require more mental effort to understand than should be necessary. I have listed some changes below but there is room to improve throughout and cleaning up (and streamlining) the writing would greatly help to improve the accessibility of the manuscript. I am aware that the main focus is on the scientific content, however the frequency of language issues distracts from this and makes the manuscript hard to follow and therefore I really think that it is worth focusing on remedying these language issues.

Specific

- Fig 2 & 3: y-axis inconsistent across columns a-d which makes it difficult to see the changes taking place. Please keep it fixed across those subplots
- Fig 2: In the decomposition rate and WSA rows, the units change from percentage to proportion between a and the other subplots
- Fig 4: why DN < -0 ? just 0
- Fig 6c: Regarding my comment on legend order: elsewhere, you consistently have red then grey then blue ("synergistic" then "no interaction" then "antagonistic") but here, in the point legend, you have synergistic then antagonistic then no net. In this way, the figure doesn't match the rest and it implies that antagonism is between synergism and no interactions which is not the case
- Fig S2 and elsewhere: Useful to show that some models "build off" others but e.g. Model 4 builds from both Model 1 and Model 2, while it is shown as just "Model 4 → Model 1". The explanation for Model 6 is also confusing, please rewrite this to better explain what is going on
- Fig S2: replace "->" with actual unicode arrows
- Fig S3: "functions by 1-7 models" → "functions by seven models"
- Fig S7: Put y-label (Number of factors) on the left hand side and put x-tick labels. Also doesn't really need the colour legend
- L40: Needs more consistent language. "Soil biota and functions" in line 40 but "soil properties and functions" in lines 42 and 45, then "soil properties and processes" in line 47, then "soil response" in line 49, then just "soil functions" in line 52. I think these all refer to decomposition rate, cellulase, and beta-glucosidase? What about the other four responses?

- L41: replace "appear" with "occur". Although later on you say they "co-act" so maybe you want to say that they "do not act in isolation"
- L45: "We chose 12 GCFs to build a factor pool" → "We created a factor pool of 12 GCFs"
- L48: "Results suggest" is a little weak. "Results show" or something would be better
- L49: "but also factor dissimilarity was key" → "but factor dissimilarity was also key"
- L59: "The effect of individual GCFs on soil properties and functions have been the focus of MANY prior studies" or something like that
- L65: "effects at a high-dimensional factor levels" → "effects at a high-dimensional factor level"
- L65: "Only few studies" → "Few studies" or "Only a few studies"
- L65: ", that is the" → "with"
- L66: " fewer than 2% of studies have explored the combined effects of three or more factors" unclear: provide context for the 2% of studies. 2% of studies studying GFC effects on soil properties and functions?
- L74: "Using this experimental approach addressed a general feature of multiple GCFs — the number of co-acting factors, which has been shown to directionally drive the effects of co-acting multiple factors on plants, soil ecosystems and the plant community" → "Using this experimental approach addressed a general feature of multiple GCFs — the number of co-acting factors — which has been shown to directionally drive the effects of co-acting multiple factors on plants, soil ecosystems and the plant community" or something because this sentence is hard to follow
- L100: "at the same time" → "simultaneously"
- L96: "The interactions between factors are defined as antagonistic when combined effects are less than expected. On the contrary, synergistic interactions cause combined effects larger than expected effects." does not flow. Replace ". On the contrary, " with ", while"
- L104: "Nevertheless, there is still a lack of established methods for measuring the interactive effects among multiple GCFs and insufficient knowledge about the potential mechanisms behind it." Not sure this sentence is needed or accurate and derails the paragraph somewhat. Maybe remove?
- L107: "examining the number of factors and dissimilarity of GCF combinations" missing a word somewhere - you're not examining these, you're examining their effects
- L108: "number of factors levels" is an awkward way of phrasing this concept - rephrase here and on L141 and L472.
- L110: there are two hypothesis "ii"s that seem to paraphrase each other. Are these the same thing or is there a missing "i"? Also I think a semicolon would work better than a full stop to separate the two/three hypotheses
- L113: "of factor interactive effects" → either "of interactive factor effects" or "of factor interactions"
- L121: "Salinity and drought had positive effects on soil pH" → did they "improve" the pH? Or you mean that they caused pH to increase (i.e. become more alkaline)?
- L122: "while decreased" → "while decreasing"
- L126: "changed directionally" ?
- L140: switch this sentence to active voice
- L144: don't start sentences with "And"
- L145: "The correlation of soil responses with the factor dissimilarity index could be caused by both factor dissimilarity effects and selecting the extreme factors that also induce a larger dissimilarity index" I don't understand this sentence

- L147: " So we cannot draw conclusions about the effect of factor dissimilarity only from the correlation analysis." don't start sentences with "So". This paragraph is confusing and seems unfinished, please look at it again
- L154: "For separating the factor identity contribution" → "To separate the contribution of factor identity"
- L163: "three soil enzymatic activity" → "three types of soil enzymatic activities" or something, this doesn't currently make sense. Also a problem elsewhere in the manuscript
- L182: " predictions(see methods)." missing a space
- L186: "The net interaction types were labeled as synergistic, antagonistic and no interaction by different colored points" don't talk about colored points without mentioning a figure. Probably just remove this sentence
- L193: "To understand the driving force of the emergence of factor interactions, we demonstrate the standardized deviations of soil responses of multiple-factor combination treatments from three null model predictions across number of factor levels and the dissimilarity range." this sentence is long and confusing
- L196 and 199: avoid saying that you "chose" best models
- L202, 205, 208 etc.: "towards synergistic interaction/effect" find another way to phrase this concept
- L217: remove "directionally" - I don't think it adds anything because you already state that combined effects become more synergistic (unless I've misunderstood something)
- L220: can you use the "i" and "ii" hypothesis numbers again here in the discussion so that we have concrete answers to the precise questions you were asking at the start, please
- L233: "the contribution of the factor identity effect is an important point that can be ignored" it is an important thing that can be ignored? What do you mean?
- L234: "The concept of "species identity (ID) effect" has been first raised in biodiversity studies for separating it from the diversity effect (DE)" I don't think either ID or DE are used going forward? Why are they given acronyms here? They're also not in the list of abbreviations in the supplementary materials
- L251: "So far, not all studies in multiple GFCs have assessed the factor identity effects, but attempts have been made by both parametric and nonparametric methods" → "While few studies investigating the effects of multiple GFCs have addressed the factor identity effect(s?), those that have done so have used both parametric and nonparametric methods (to do...)" or something. This sentence is just a bit awkward, rework it
- L371: "differ in nature" is vague
- L372: "effect mechanisms, namely salinity" → "effect mechanisms: salinity"
- L390: "Then additional" → "Then an additional"
- L391: " added to every 150 mL cup" missing a "." after "cup"
- L411: "and four soil extracellular enzyme activity" → "and the activity of four extracellular soil enzymes"
- L452: "The distributions of normalized factor dissimilarity indices in three number of factor groups" - "three number of" doesn't make sense
- L463: "In ecology studies" → "In ecology" or "In ecological studies"
- L464: "For commonly-used null models, the additive model assumes the joint effect of multiple factors will be the sum of the effects of single factors; the multiplicative model assumes the effects

of single factors are combined by proportional change. In the dominative model, the factor with largest absolute effect overrides other factors". Why is this split over two sentences? Is additive being singled out for a reason? Also would be good to mention the implications of each null model - negatively-correlated/positively-correlated/non-correlated sensitivity

- L539: " we applied a framework for ecological stressor interaction measuring" → " we applied a framework for measuring ecological stressor interactions"

- L544: "For positive responses to multiple GFCs" → "For variables with positive responses to multiple GFCs" or something

- L547: "but as antagonistic interactions when" → "and as antagonistic interactions when"

- L759: Breiman, L. reference missing title - it seems like it should be "Random forests"

- Supplementary L23: "Drought has been taken as a major global threat" → "Drought is a major global threat to X"

- Supplementary L85: "The existence of Lithium" → "The presence of lithium"

- Supplementary L105: "inserted to a sieving machine" → "inserted into a sieving machine"

- Supplementary L108: "weighted" → "weighed"

- Supplementary L112: why are there square brackets around everything here? It also doesn't seem like this would yield a percentage, but rather a proportion. Missing a multiplication by 100?

- Supplementary L115: "(Lipton,the United Kingdom)" → missing a space after the comma and also just put "United Kingdom" without the "the"

- Supplementary L143: Missing a space after the "."

- Supplementary overall: Model 6 is confusing to understand overall - Table S6 shows only one model construction but later it becomes two. Please make this clearer

- Supplementary table 1: why are they "stressors" now and not "GFCs"?

- Supplementary table 3: capitalise "Water stable soil aggregation"

- Supplementary table 4: capitalise "cellulase" and "phosphatase"

- Supplementary table 6: "formular" → "formula"

- Throughout: "P value" inconsistent - sometimes hyphen, sometimes capitalised. Pick one (probably the "P value" format, since it seems to be used more in Nature journals)

Reviewer #3:

Remarks to the Author:

The authors have strongly improved the manuscript and convincingly tackled all reviewer comments.

I have only a few comments that may improve the manuscript/fix errors.

I generally like the hierarchical statistical framework. However, it remains unclear why two methods should be used. When the authors claim that they provide a „practical method to evaluate the contribution“ then it should be clear which methods (GLM or RF) should be preferred.

In fact, the ideal method would depend on the data distribution. I would suggest that they move of the methods in the SI as the results are largely the same and then refer to this method there. This would also make the manuscript more succinct.

General: You introduce the acronym GLM, but then use a couple of types the written out version

432 the „vegan“ package

Supporting Information

290 Should be „indices co-vary with“ - remove „are“

297 „taking into account“

Reviewer Comments:

Reviewer #1 (Remarks to the Author):

General Thoughts

- I am mostly pleased with the revisions, and I am also happy to see that the authors have removed the groupings of factor dissimilarity in favour of a continuous gradient.
- My main issues currently are to do with the language and presentation/flow of the manuscript. The authors with stronger English skills need to take another look over the text, as many parts are awkwardly-written and require more mental effort to understand than should be necessary. I have listed some changes below but there is room to improve throughout and cleaning up (and streamlining) the writing would greatly help to improve the accessibility of the manuscript. I am aware that the main focus is on the scientific content, however the frequency of language issues distracts from this and makes the manuscript hard to follow and therefore I really think that it is worth focusing on remedying these language issues.

>>RESPONSE. Thank you very much for your positive assessment. We apologize for the language mistakes and occasional lack of clarity. We have worked to improve the language of the manuscript. We are very grateful to the reviewer for going the extra mile to provide such detailed language suggestions, which we have all taken into consideration (please see detailed comments below).

- Fig 2 & 3: y-axis inconsistent across columns a-d which makes it difficult to see the changes taking place. Please keep it fixed across those subplots
- Fig 2: In the decomposition rate and WSA rows, the units change from percentage to proportion between a and the other subplots

>>RESPONSE. Thanks, we have made those changes.

- Fig 4: why DN < -0 ? just 0

- Fig 6c: Regarding my comment on legend order: elsewhere, you consistently have red then grey then blue ("synergistic" then "no interaction" then "antagonistic") but here, in the point legend, you have synergistic then antagonistic then no net. In this way, the figure doesn't match the rest and it implies that antagonism is between synergism and no interactions which is not the case

>>RESPONSE. Thanks very much. We agree with all points, and we have made all the proposed changes.

- Fig S2 and elsewhere: Useful to show that some models "build off" others but e.g. Model 4 builds from both Model 1 and Model 2, while it is shown as just "Model 4 → Model 1". The explanation for Model 6 is also confusing, please rewrite this to better explain what is going on - Fig S2: replace "->" with actual unicode arrows

>>RESPONSE. Thanks, we agree. We added more explanations for Model 4, 5 and 6 in Figure S2 a, and we have replaced the "->" with arrows.

b Testing hypothesis by comparing models:

- Model 1:** Variability of soil response to multiple GCFs explained by three null model predictions, which is regarded as the factor identity effect.
- Model 2:** Variability of soil responses to multiple GCFs explained by number of factor effect.
- Model 3:** Variability of soil responses to multiple GCFs explained by factor dissimilarity indices.
- Model 4 → Model 1:** Variability of soil responses to multiple GCFs further explained by number of factor effect on the basis of including factor identity effect.
- Model 5 → Model 1:** Variability of soil responses to multiple GCFs further explained by factor dissimilarity effect on the basis of including factor identity effect.
- Model 6 → Model 4:** Variability of soil responses to multiple GCFs further explained by factor dissimilarity effect on the basis of including factor identity effect and number of factor effect.
- Model 6 → Model 5:** Variability of soil responses to multiple GCFs further explained by number of factor effect on the basis of including factor identity effect and factor dissimilarity effect.
- Model 7 → Model 6:** Variability of soil responses to multiple GCFs further explained by factor composition information on the basis of including all hypothetical predictors and factor identity effect.

- Fig S3: "functions by 1-7 models" → "functions by seven models"

- Fig S7: Put y-label (Number of factors) on the left hand side and put x-tick labels. Also doesn't really need the colour legend

>>RESPONSE. Thank you for those comments. We have made all proposed changes.

- L40: Needs more consistent language. "Soil biota and functions" in line 40 but "soil properties and functions" in lines 42 and 45, then "soil properties and processes" in line 47, then "soil response" in line 49, then just "soil functions" in line 52. I think these all refer to decomposition rate, cellulase, and beta-glucosidase? What about the other four responses?

>>RESPONSE. Thank you for this comment. We think "Soil biota and functions" in line 40 refers to not only the decomposition rate, cellulase and beta-glucosidase, but also other responses of soil ecosystems that can be affected by anthropogenic factors. It is a general statement about the potential risks of anthropogenic factors on soil ecosystems. For consideration of consistency, we changed the "soil properties and processes" in line 47 to "soil properties and functions". In line 49 we changed the sentence to "By analyzing deviations of soil properties and functions from three null model predictions,(...)." In line 52, "soil functions" refers to decomposition rate, cellulase and beta-glucosidase, "soil properties" refers to soil pH and water stable soil aggregates.

- L41: replace "appear" with "occur". Although later on you say they "co-act" so maybe you want to say that they "do not act in isolation"

>>RESPONSE. Thanks, we replaced "appear" with "occur". Yes, we say "co-act" because we want to say that they do not act in isolation.

- L45: "We chose 12 GCFs to build a factor pool" → "We created a factor pool of 12 GCFs"

- L48: "Results suggest" is a little weak. "Results show" or something would be better

- L49: "but also factor dissimilarity was key" → "but factor dissimilarity was also key"

>>RESPONSE. Thanks, we made those changes.

- L59: "The effect of individual GCFs on soil properties and functions have been the focus of MANY prior studies" or something like that

>>RESPONSE. Thanks, we added "many".

- L65: "effects at a high-dimensional factor levels" → "effects at a high-dimensional factor level"

- L65: "Only few studies" → "Few studies" or "Only a few studies"

- L65: ", that is the" → "with"

>>RESPONSE. Thanks, we made those changes. The sentence now reads: "Only a few studies have addressed the effects at a high-dimensional factor level with concurrent effects of a larger number of factors.".

- L66: " fewer than 2% of studies have explored the combined effects of three or more factors" unclear: provide context for the 2% of studies. 2% of studies studying GFC effects on soil properties and functions?

>>RESPONSE. Thanks, according to the reference, we modified the sentence as: "*fewer than 2% of experimental studies have explored the combined effects of three or more factors in the context of soils.*"

- L74: "Using this experimental approach addressed a general feature of multiple GCFs — the number of co-acting factors, which has been shown to directionally drive the effects of co-acting multiple factors on plants, soil ecosystems and the plant community" → "Using this experimental approach addressed a general feature of multiple GCFs — the number of co-acting factors — which has been shown to directionally drive the effects of co-acting multiple factors on plants, soil ecosystems and the plant community" or something because this sentence is hard to follow

- L100: "at the same time" → "simultaneously"

- L96: "The interactions between factors are defined as antagonistic when combined effects are less than expected. On the contrary, synergistic interactions cause combined effects larger than expected effects." does not flow. Replace ". On the contrary, " with ", while"

>>RESPONSE. Thanks, we made those changes.

- L104: "Nevertheless, there is still a lack of established methods for measuring the interactive effects among multiple GCFs and insufficient knowledge about the potential mechanisms behind it." Not sure this sentence is needed or accurate and derails the paragraph somewhat. Maybe remove?

>>RESPONSE. Thanks, we changed this sentence to "*Nevertheless, there is still a lack of established methods for identifying the net interactions for multiple co-acting GCFs and insufficient knowledge about the potential mechanisms underpinning such effects.*". This is more consistent with the second last sentence.

- L107: "examining the number of factors and dissimilarity of GCF combinations" missing a word somewhere - you're not examining these, you're examining their effects

>>RESPONSE. Thanks, we added "*effects of the number of factors*" in the sentence.

- L108: "number of factors levels" is an awkward way of phrasing this concept - rephrase here and on L141 and L472.

>>RESPONSE. Thanks, we changed the sentence to "*a microcosm experiment (Fig.1) with 2, 5 and 8 factor levels to (...).*". And also the concept in other places.

- L110: there are two hypothesis "ii"s that seem to paraphrase each other. Are these the same thing or is there a missing "i"? Also I think a semicolon would work better than a full stop to separate the two/three hypotheses

- L113: "of factor interactive effects" → either "of interactive factor effects" or "of factor interactions"

>>RESPONSE. Thanks, we choose "*factor interactions*".

- L121: "Salinity and drought had positive effects on soil pH" → did they "improve" the pH? Or you mean that they caused pH to increase (i.e. become more alkaline)?

>>RESPONSE. Yes, thank you. We revised to "Salinity and drought caused soil pH to increase"

- L122: "while decreased" → "while decreasing"

- L126: "changed directionally" ?

- L140: switch this sentence to active voice

- L144: don't start sentences with "And"

>>RESPONSE. Thanks; we have made all these changes.

- L145: "The correlation of soil responses with the factor dissimilarity index could be caused by both factor dissimilarity effects and selecting the extreme factors that also induce a larger dissimilarity index" I don't understand this sentence

- L147: " So we cannot draw conclusions about the effect of factor dissimilarity only from the correlation analysis." don't start sentences with "So". This paragraph is confusing and seems unfinished, please look at it again

>>RESPONSE. Thanks for this comment. We modified the sentence to: "*However, the correlation of soil responses with the factor dissimilarity index could be caused by the artefact that factor combinations with larger dissimilarity indices also have a higher chance of including the factors with extreme effect size. Therefore, only the correlation analysis by itself is insufficient for evaluating the real effects of factor dissimilarity*".

- L154: "For separating the factor identity contribution" → "To separate the contribution of factor identity"

- L163: "three soil enzymatic activity" → "three types of soil enzymatic activities" or something, this doesn't currently make sense. Also a problem elsewhere in the manuscript

>>RESPONSE. Thanks, we made all these changes.

- L182: " predictions(see methods)." missing a space

- L186: "The net interaction types were labeled as synergistic, antagonistic and no interaction by different colored points" don't talk about colored points without mentioning a figure. Probably just remove this sentence

>>RESPONSE. Thanks, we have made those changes.

- L193: "To understand the driving force of the emergence of factor interactions, we demonstrate the standardized deviations of soil responses of multiple-factor combination treatments from three null model predictions across number of factor levels and the dissimilarity range." this sentence is long and confusing

>>RESPONSE. Thanks, we have worked on it and modified the sentence to: *"To evaluate the drivers of factor interactions, we assessed the standardized deviations of soil responses from three null model predictions across the dissimilarity range and three factor levels, respectively."*

- L196 and 199: avoid saying that you "chose" best models

>>RESPONSE. Thanks, we modified the sentence to: *"From the three null models, the model with the smallest sum of squared deviations is used for estimating the deviation for each soil response. For soil decomposition rate, the additive model has the smallest SSD, while the dominative models have the smallest SSD for soil pH and water-stable soil aggregates. For soil enzymatic activity, the multiplicative models have the smallest SSD, except for cellulase activity (dominative model) (Supplementary Table. 9)."*

- L202, 205, 208 etc.: "towards synergistic interaction/effect" find another way to phrase this concept

>>RESPONSE. Thanks for this comment. We modified the sentence to: *"The standardized deviation of soil decomposition rate from additive model predictions is correlated with the factor dissimilarity index, with synergistic interactions becoming more frequent with higher dissimilarity (Fig. 5). The standardized deviation of cellulase and β -glucosidase activity from the null models with the smallest SSD also show correlations with factor dissimilarity index, with more frequent synergistic interactions appearing with higher dissimilarity (Fig. 6). The soil decomposition rate responses to eight-factor treatments deviate significantly from the additive model (Supplementary Fig. 4), and also the soil enzymatic activity responses to higher number of factor treatments show significant deviation from the null models with the smallest SSD (Supplementary Fig. 5). These results suggest that the joint GCF effects on soil properties and functions deviate from null model predictions, and the number of factors and factor dissimilarity may drive the occurrence of more synergistic factor interactions"*.

- L217: remove "directionally" - I don't think it adds anything because you already state that combined effects become more synergistic (unless I've misunderstood something)

>>RESPONSE. We agreed and made the change.

- L220: can you use the "i" and "ii" hypothesis numbers again here in the discussion so that we have concrete answers to the precise questions you were asking at the start, please

>>RESPONSE. Thanks, we modified the sentence as: *"As hypothesized, (i) the effects of factor dissimilarity played an important role in predicting the variability of soil responses to multiple GCFs; (ii) a larger dissimilarity among factors or larger number of factors will cause greater deviation of joint effects on soil properties and functions from null models; (iii) co-acting factors with higher dissimilarity tend to have more synergistic interactions."*

- L233: "the contribution of the factor identity effect is an important point that can be ignored" it is an important thing that can be ignored? What do you mean?

>>RESPONSE. Thanks for this comment. Sorry the original sentence was misleading, here we changed the sentence to: *"However, the contribution of the factor identity effect is*

an important point that cannot be ignored, otherwise this may result in contradictory results.”.

- L234: "The concept of 'species identity (ID) effect' has been first raised in biodiversity studies for separating it from the diversity effect (DE)" I don't think either ID or DE are used going forward? Why are they given acronyms here? They're also not in the list of abbreviations in the supplementary materials

>>RESPONSE. Thanks, this is true; we removed those abbreviations.

- L251: "So far, not all studies in multiple GFCs have assessed the factor identity effects, but attempts have been made by both parametric and nonparametric methods" "While few studies investigating the effects of multiple GFCs have addressed the factor identity effect(s?), those that have done so have used both parametric and nonparametric methods (to do...)" or something. This sentence is just a bit awkward, rework it

>>RESPONSE. Thanks, we modified the sentence to: "Although only a few studies on multiple GFCs have assessed factor identity effects, attempts have been made in some studies to use both parametric and nonparametric methods.”.

- L371: "differ in nature" is vague

>>RESPONSE. We agree; we have changed it to "that differ in their nature (physical, chemical, biological)."

- L372: "effect mechanisms, namely salinity" "effect mechanisms: salinity"

- L390: "Then additional" "Then an additional"

- L391: " added to every 150 mL cup" missing a "." after "cup"

- L411: "and four soil extracellular enzyme activity" "and the activity of four extracellular soil enzymes"

>>RESPONSE. Thanks, we made those changes.

- L452: "The distributions of normalized factor dissimilarity indices in three number of factor groups" - "three number of" doesn't make sense

>>RESPONSE. We changed it to: "The distributions of normalized factor dissimilarity indices in three factor levels and (...)."

- L463: "In ecology studies" "In ecology" or "In ecological studies"

>>RESPONSE. Thanks, we corrected it to "In ecological studies”.

- L464: "For commonly-used null models, the additive model assumes the joint effect of multiple factors will be the sum of the effects of single factors; the multiplicative model assumes the effects of single factors are combined by proportional change. In the dominative model, the factor with largest absolute effect overrides other factors". Why is this split over two sentences? Is additive

being singled out for a reason? Also would be good to mention the implications of each null model
- negatively-correlated/positively-correlated/non-correlated sensitivity

>>RESPONSE. Thank you for this comment, we think using a semicolon here was misleading. We also appreciate the suggestion of adding implications of sensitivity correlations. We modified the sentences to: *"For commonly-used null models, the additive model assumes that the joint effect of multiple factors will be the sum of the effects of the single factors, indicating that the sensitivities of the target to factors are negatively correlated. The multiplicative model assumes that the effects of single factors are combined by proportional change, meaning that the factor sensitivities are non-correlated. In the dominative model, the factor with the largest absolute effect overrides other factors, implying the factor sensitivities are positively correlated"*.

- L539: " we applied a framework for ecological stressor interaction measuring" → " we applied a framework for measuring ecological stressor interactions"
- L544: "For positive responses to multiple GFCs" → "For variables with positive responses to multiple GFCs" or something
- L547: "but as antagonistic interactions when" → "and as antagonistic interactions when"
- L759: Breiman, L. reference missing title - it seems like it should be "Random forests"

>>RESPONSE. Thanks, we have made those changes.

- Supplementary L23: "Drought has been taken as a major global threat" → "Drought is a major global threat to X"
- Supplementary L85: "The existence of Lithium" → "The presence of lithium"
- Supplementary L105: "inserted to a sieving machine" → "inserted into a sieving machine"
- Supplementary L108: "weighted" → "weighed"

>>RESPONSE. Thanks, we have made those changes.

- Supplementary L112: why are there square brackets around everything here? It also doesn't seem like this would yield a percentage, but rather a proportion. Missing a multiplication by 100?

>>RESPONSE. Thanks for this comments, we have revised the formula as:

$$WSA (\%) = (dry\ matter - coarse\ matter) / (sample\ weight - coarse\ matter) \times 100 (1)$$
.

- Supplementary L115: "(Lipton,the United Kingdom)" → missing a space after the comma and also just put "United Kingdom" without the "the"
- Supplementary L143: Missing a space after the "."

>>RESPONSE. Thanks, we have made those changes.

- Supplementary overall: Model 6 is confusing to understand overall -
Table S6 shows only one model construction but later it becomes two. Please make this clearer

>>RESPONSE. Thanks for this comment. We have worked on making the model explanation more clear by adding more explanations to Figure S2. Each model only has one construction, but later in Table S7 and S8, we compared Model 6 with the other two different models

(Model 4 and 5). That is why there are two rows in those Tables to indicate the result of the comparison with Model 6.

- Supplementary table 1: why are they "stressors" now and not "GFCs"?

>>RESPONSE. Thanks, we have changed it to GFCs.

- Supplementary table 3: capitalise "Water stable soil aggregation"

- Supplementary table 4: capitalise "cellulase" and "phosphatase"

- Supplementary table 6: "formular" → "formula"

- Throughout: "P value" inconsistent - sometimes hyphen, sometimes capitalised. Pick one (probably the "P value" format, since it seems to be used more in Nature journals)

>>RESPONSE. Thanks, we have made those changes.

Reviewer #3 (Remarks to the Author):

The authors have strongly improved the manuscript and convincingly tackled all reviewer comments. I have only a few comments that may improve the manuscript/fix errors.

>>RESPONSE. Thank you very much for your positive assessment.

I generally like the hierarchical statistical framework. However, it remains unclear why two methods should be used. When the authors claim that they provide a "practical method to evaluate the contribution"; then it should be clear which methods (GLM or RF) should be preferred. In fact, the ideal method would depend on the data distribution. I would suggest that they move one of the methods in the SI as the results are largely the same and then refer to this method there. This would also make the manuscript more succinct.

>>RESPONSE. Thank you for this comment. We added the hierarchical model based on the GLM algorithm because one of the other reviewers had the concern that only the R^2 increment of the RF model cannot support that adding the predictor improved model performance. Thus we included GLM with AIC evaluations for a proper model selection process. We admit that the two models being used here are a bit confusing and lengthy, but we think we could not make a recommendation, at least at the current stage, which model should be preferred. Both RF and GLM models have advantages and limitations, and often both results are given also in other studies. Considering succinctness, we modified the text in the Results to: *"The hierarchical modeling based on the GLM algorithm also showed similar results as the RF models (Supplementary Table 7 and 8), Collectively, both machine learning and GLM algorithms indicate that the number of factors and factor dissimilarity are important predictors for the variability of soil responses to multiple GFCs"*.

General: You introduce the acronym GLM, but then use a couple of types the written out version

>>RESPONSE. Thanks. We have corrected them.

432 the "vegan" package

>>RESPONSE. Thanks. We have corrected it.

Supporting Information

290 Should be "indices co-vary with"; - remove
"are";

297 "taking into account";

>>RESPONSE. Thanks. We have made those changes.

Reviewers' Comments:

Reviewer #1:

Remarks to the Author:

General comments

- I am mostly satisfied with the changes made and I think that the manuscript is a valuable and well-executed contribution
- The last few minor things are mainly language and clarity again, since it would be a shame to lose readers due to convoluted text when the results are strong
- I had a brief look at the code, however, on https://github.com/mohanb96/Effect_of_multiple_GCF_dissimilarity/blob/main/FSE_Code_summary.R and, if you want it to be a useable resource for people to be able to review and replicate, this needs to be cleaned up. I have included these comments in "Remarks on code availability" section of the review but I'm not sure if Nature passes these on to you so I repeat them here just in case
- Firstly, 2500 lines is too long - this would be better split into several files with their own roles which are then called by some "master" file so that the different processes and steps are compartmentalised and understandable.
- Much of the code is very heavy on copy and pasting the same line and then modifying the number at the end which is quite a risky way to operate given the likelihood of a typo somewhere.
- There's also lots of hard-coded stuff like `N=1:123` instead of `N=1:nrow(sf)` which is quite sensitive to anything changing between versions and relies on the authors making sure that they clean up every single instance of an altered variable or input.
- There are some incredibly deeply nested data structures which are accessed with e.g. ``length(which(Diviation_3_models[[i_response]][[n_model]][Diviation_3_models[[i_response]][[n_model]][["Lv"]]==Lv,"interaction_type"]=="Synergistic"))`` which is complicated to follow and therefore more likely to contain an error somewhere. The tidyverse-style workflow used in some places is a lot easier to follow and therefore easier to find errors in
- PS "Diviation" -> "Deviation" I think, which should be solvable with just a find and replace, although obviously spelling mistakes in code aren't a huge deal as long as they're consistent
- The authors also repeatedly overwrite globally-defined variables with new values (e.g. "c <- f2_dis\$dissim" and "c<-f5_dis\$dissim" etc. Also please avoid using "c" as a variable name since in R it is also a very important base function, and, while overloading it to also be a variable is possible, I do not think it's advisable), increasing the likelihood that some variable further up in the file gets left out and introduces problems later on.
- I don't have reason to suspect that anything is wrong with the code, given that the actual figures and results make sense, but it would be difficult to actually show that the code is working as intended in its current state.

Specific comments

- L108 and 111: don't capitalise "Factor"
- L118: redundancy in "However, according to significance tests based on adjusted P values (n = 8), none of the 12 single factors had significant effects...". Probably just start with "However, none of

the 12 single factors had significant effects..."

- L146: "Therefore, only the correlation analysis by itself is insufficient for evaluating the real effect of factor dissimilarity." -> "Therefore, the correlation analysis alone is insufficient to evaluate the true effect of factor dissimilarity."

- L227: "However, the contribution of the factor identity effect is an important point that cannot be ignored, otherwise this may result in contradictory results" -> "However, the factor identity effect cannot be ignored, as it may drive contradictory results" or something like that, the sentence is awkward

- L271: "And also, water-stable soil aggregation is negatively correlated" -> "Water-stable soil aggregation is also negatively correlated". Also why "water-stable soil aggregation" here but "WSA" 2 lines down

- L276: "As we saw contradictory results by analyzing data with or without considering the factor identity effect, to avoid misinterpreting results, we here suggest separating factor identity effects from other effects driven by multiple GCFs." -> "Because the factor identity effect caused contradictory results, we suggest separating factor identity effects from other effects driven by multiple GCFs." or something

- L285: "Our study has found there is more emergence of synergistic factor interaction on..." -> "We found increased emergence of synergistic interactions on..."

- L298: "Factor direct interactions usually amplify the intensity of single factors. Thus, the emergence..." -> "Direct factor interactions usually amplify the intensity of single factors and, thus, the emergence...". Also is this true? This sounds like the default net effect of multiple GCFs is synergistic and -- quoting a meta-analysis from Côté et al 2016 -- synergies are not the most prevalent type of interaction. Is this different in soil ecosystems or when taking newer literature into account?

- L336: "The experiment was set up with a GCF pool that includes 12 factors: ... (PFAS) and lithium" in a long list like this I would recommend putting a comma before "lithium"

- L355: "with a sandy loamy texture" needs more specifics on what this signifies for non soil-ecologists

- L357: "2 mm sieve to remove large stones and big grass roots" what does this leave intact? what things <2mm are you deliberately keeping or is this just the minimum workable sieve width with soil?

- L358: "under 121°C" -> "at 121°C"

- L360: "it was sterilized to avoid large local effects on soil microbes." what do you mean?

- L367: paragraph starting "This study focuses on 12 frequently occurring GCFs" this paragraph is redundant as you have already stated this in L334

- L372: This section overlaps with the previous section on "Soil preparation and incubation system". Suggest merging these two sections to avoid the feeling of going back and forth

- L408: "pH, water-stable aggregates, and the activity" include acronym of WSA here, like you did for PFAS in the GCF description paragraph

- L440 (equation): maybe it's just my pdf version but the summation seems to be missing a superscript, check this

- L550: "functions)⁴⁸," there is a comma in your superscript instead of after it (at least in my pdf version)

- Figs 2 and 3 captions: "a(1) to e(3)" and "a(1) to e(4)" this little preface thing is not needed (unless

the journal specifically asked for it?)

- Figs 2 and 3: you have addressed the y-axes being inconsistent for columns b-d but not column a. a-d are all showing the same response variable for each row, and should have have the same scale to be comparable

- Figs 5 and 6: I missed this the other times so apologies for the confusion, but like mentioned before, for fig 4, can you put the grey dots (no interaction) between the red and blue dots in the legends? It just gives the feeling of counting like "1, 3, 2" in its current form

- Supp tab 1, you changed stressors to GCFs in the legend but not the actual table

- Throughout: avoid mixing past and present tenses in sentences

- Throughout: please look through for cases where you have passive voice and switch to active, where possible. You have a mixture of both and, since the active voice is much easier to follow, I would favour that (especially in the methods section which is description-heavy and alternates between the two styles)

- I am mostly satisfied with the changes made and I think that the manuscript is a valuable and well-executed contribution
- The last few minor things are mainly language and clarity again, since it would be a shame to lose readers due to convoluted text when the results are strong
- I had a brief look at the code, however, on https://github.com/mohanb96/Effect_of_multiple_GCF_dissimilarity/blob/main/FSE_Code_summary.R and, if you want it to be a useable resource for people to be able to review and replicate, this needs to be cleaned up. I have included these comments in "Remarks on code availability" section of the review but I'm not sure if Nature passes these on to you so I repeat them here just in case.
- Firstly, 2500 lines is too long - this would be better split into several files with their own roles which are then called by some "master" file so that the different processes and steps are compartmentalised and understandable.

>>RESPONSE. Thank you for your thoughtful feedback and for taking the time to review the revisions to our manuscript. We greatly appreciate your recognition of the changes made and are glad to hear that you find the manuscript to be a valuable and well-executed contribution.

We understand your concerns regarding the language and clarity of the text. We will carefully address the minor issues you mentioned to ensure that the writing is concise and accessible.

Regarding the code available at the GitHub repository, we appreciate your suggestion for cleaning it up. We share your commitment to providing a usable resource for others and will take steps to enhance its organization and clarity, making it easier for readers to review and replicate our work. Thank you for including additional comments in the "Remarks on code availability" section; please see there for our responses.

Specific comments

- L108 and 111: don't capitalise "Factor"
- L118: redundancy in "However, according to significance tests based on adjusted P values (n = 8), none of the 12 single factors had significant effects...". Probably just start with "However, none of the 12 single factors had significant effects..."

>>RESPONSE. Thanks, we have made those changes.

- L146: "Therefore, only the correlation analysis by itself is insufficient for evaluating the real effect of factor dissimilarity." -> "Therefore, the correlation analysis alone is insufficient to evaluate the true effect of factor dissimilarity."

- L227: "However, the contribution of the factor identity effect is an important point that cannot be ignored, otherwise this may result in contradictory results" -> "However, the factor identity effect cannot be ignored, as it may drive contradictory results" or something like that, the sentence is awkward

>>RESPONSE. Thank you for the suggestions, we have modified the sentences.

- L271: "And also, water-stable soil aggregation is negatively correlated"-> "Water-stable soil aggregation is also negatively correlated". Also why "water-stable soil aggregation" here but "WSA" 2 lines down

>>RESPONSE. Thanks, we have modified the sentence to: "*WSA is also negatively correlated (...)*". We provide the whole name of the abbreviation for water-stable soil aggregates at L123, as here is the first time it appears in the article. We checked all the following "water-stable soil aggregates" and changed it to "WSA".

- L276: "As we saw contradictory results by analyzing data with or without considering the factor identity effect, to avoid misinterpreting results, we here suggest separating factor identity effects from other effects driven by multiple GCFs." -> "Because the factor identity effect caused contradictory results, we suggest separating factor identity effects from other effects driven by multiple GCFs." or something

>>RESPONSE. Thanks, we have modified the sentence as you suggested.

- L285: "Our study has found there is more emergence of synergistic factor interaction on..." -> "We found increased emergence of synergistic interactions on..."

>>RESPONSE. Thanks, we have modified the sentence.

- L298: "Factor direct interactions usually amplify the intensity of single factors. Thus, the emergence..." -> "Direct factor interactions usually amplify the intensity of single factors and, thus, the emergence...". Also is this true? This sounds like the default net effect of multiple GCFs is synergistic and -- quoting a meta-analysis from Côté et al 2016 -- synergies are not the most prevalent type of interaction. Is this different in soil ecosystems or when taking newer literature into account?

>>RESPONSE. We appreciate your suggestion. We revised it to "*Direct factor interactions usually amplify the intensity of single factors and, thus, the emergence...*". Regarding your concern about the assumption that the default net effect of multiple global change factors (GCFs) is synergistic, we don't think it is true. The information we want to deliver here is, direct factor interactions can amplify the intensity of single factors, it might be one way that contributes to the synergistic interactions we observed in the study. Amplified single factor intensity doesn't mean synergistic interaction effect.

- L336: "The experiment was set up with a GCF pool that includes 12 factors: ... (PFAS) and lithium" in a long list like this I would recommend putting a comma before "lithium"

>>RESPONSE. Thanks, we have made the change.

- L355: "with a sandy loamy texture" needs more specifics on what this signifies for non soil-ecologists

>>RESPONSE. Thank you for your suggestion. A sandy loamy texture refers to soil that contains a balanced proportion of sand, silt, and clay particles, with sand being the predominant component.

- L357: "2 mm sieve to remove large stones and big grass roots" what does this leave intact? what things <2mm are you deliberately keeping or is this just the minimum workable sieve width with soil?

>>RESPONSE. Thank you for your comment. The use of a 2 mm sieve in soil preparation is a standard practice aimed at achieving a balance between removing unwanted large debris while retaining the essential finer components of the soil. The 2 mm sieve is selected to remove large stones and substantial plant material such as big grass roots, which could interfere with subsequent soil experiments. By removing these larger items, we ensure a more uniform substrate for study, minimizing physical obstructions that could affect experimental outcomes or equipment.

- L358: "under 121°C" -> "at 121°C"

>>RESPONSE. Thanks, we have made the change.

- L360: "it was sterilized to avoid large local effects on soil microbes." what do you mean?

>>RESPONSE. Thanks, we have changed the sentence to: *"it was sterilized to avoid large local effects of concentrated chemicals on soil microbes."*

- L367: paragraph starting "This study focuses on 12 frequently occurring GCFs" this paragraph is redundant as you have already stated this in L334

>>RESPONSE. Thanks, we have removed this paragraph.

- L372: This section overlaps with the previous section on "Soil preparation and incubation system". Suggest merging these two sections to avoid the feeling of going back and forth

>>RESPONSE. Thanks, we have combined the two sections into one section: "Soil preparation and GCF implementation".

- L408: "pH, water-stable aggregates, and the activity" include acronym of WSA here, like you did for PFAS in the GCF description paragraph

>>RESPONSE.Thanks for pointing out. We have changed "water-stable aggregates" to "WSA".

- L440 (equation): maybe it's just my pdf version but the summation seems to be missing a superscript, check this

>>RESPONSE. Thanks for this comment. When the index set (for j) is implied, the superscript is not always needed. For a more formal mathematical writing, we modified the equation to: " $\square_{\square} = \sum_{\square \in \square} \square_{\square}$ ".

- L550: "functions)⁴⁸," there is a comma in your superscript instead of after it (at least in my pdf version)

>>RESPONSE. Sorry, we could not find it.

- Figs 2 and 3 captions: "a(1) to e(3)" and "a(1) to e(4)" this little preface thing is not needed (unless the journal specifically asked for it?)

>>RESPONSE. Thanks for the suggestion. We have removed them.

- Figs 2 and 3: you have addressed the y-axes being inconsistent for columns b-d but not column a. a-d are all showing the same response variable for each row, and should have have the same scale to be comparable

>>RESPONSE. Thank you for your feedback regarding Figures 2 and 3. We appreciate your attention to the consistency of the y-axes across the columns. While we understand your concern about the comparability of the response variable across all columns (a-d), we intentionally chose to use y-axis scales for column (b-d) to zoom in the variation range shared only by multiple factor treatments. y-axis scales for column a are better for capturing the overall picture of all single and multiple treatments. We believe that the slight difference in y-axis scales will not hinder the delivery of the information, and it allows for a clearer representation of the distinct trends presented in each column.

- Figs 5 and 6: I missed this the other times so apologies for the confusion, but like mentioned before, for fig 4, can you put the grey dots (no interaction) between the red and blue dots in the legends? It just gives the feeling of counting like "1, 3, 2" in its current form

>>RESPONSE. Thanks, We have changed them to "1,2,3" order.

- Supp tab 1, you changed stressors to GCFs in the legend but not the actual table

>>RESPONSE. Thanks. We have changed it.

- Throughout: avoid mixing past and present tenses in sentences

- Throughout: please look through for cases where you have passive voice and switch to active, where possible. You have a mixture of both and, since the active voice is much easier to follow, I would favour that (especially in the methods section which is description-heavy and alternates between the two styles)

>>RESPONSE. Thanks for your advice. We have worked on it and made it clear and easier to read.

Reviewer #1 (Remarks on code availability):

I have had a brief look at the code but the file is 2500 lines long and would take a very long time to properly evaluate. This is why I have said "No" in response to "Have you reviewed the code?" despite having comments.

Firstly, 2500 lines is too long - this would be better split into several files with their own roles which are then called by some "master" file so that the different processes and steps are compartmentalised and understandable.

Much of the code is very heavy on copy and pasting the same line and then modifying the number at the end which is quite a risky way to operate given the likelihood of a typo somewhere.

```
>>RESPONSE. Thank you for your constructive comments. We agreed and splitted the 2500 lines into several more manageable files that would significantly enhance readability and maintainability. We have separated the code into eight files:
"Part_1_dsissimilarity_calculation.R", "Part_2_Data_Preparation.R",
"Part_3_Correlation_RandomForest_Analysis.R", "Part_4_Heirarchical_analysis.R",
"Part_5_Net_Interaction_Analysis.R", "Functions_for_Part_1.R",
"Functions_for_Part_3.R", "Functions_for_Part_5.R".
```

There's also lots of hard-coded stuff like `N=1:123` instead of `N=1:nrow(sf)` which is quite sensitive to anything changing between versions and relies on the authors making sure that they clean up every single instance of an altered variable or input.

```
>>RESPONSE. Thanks for this comment. We have changed them to N=1:nrow(xx).
```

There are some incredibly deeply nested data structures which are accessed with e.g.

```
`length(which(Diviation_3_models[[i_response]][[n_model]][Diviation_3_models[[i_response]][[n_model]][["Lv"]]==Lv,"interaction_type"]=="Synergistic"))`
```

which is complicated to follow and therefore more likely to contain an error somewhere.

```
>>RESPONSE. Thanks, we have worked to make the code lines shorter.
```

- PS "Diviation" -> "Deviation" I think, which should be solvable with just a find and replace, although obviously spelling mistakes in code aren't a huge deal as long as they're consistent

```
>>RESPONSE. Thanks, we have corrected them.
```

The authors also repeatedly overwrite globally-defined variables with new values (e.g. "c<-f2_dis\$dissim" and "c<-f5_dis\$dissim" etc. Also please avoid using "c" as a variable name since in R it is also a very important base function, and, while overloading it to also be a variable is possible, I do not think it's advisable), increasing the likelihood that some variable further up in the file gets left out and introduces problems later on.

```
>>RESPONSE. Thanks, we have changed them.
```

I don't have reason to suspect that anything is wrong with the code given that the actual figures and results seem to make sense, but it would be difficult to actually show that the code is working as intended in its current state.